# Nivolumab plus chemoradiotherapy in locally-advanced cervical cancer: the NICOL phase 1 trial

Concurrent chemoradiotherapy (CRT) with blockade of the PD-1 pathway may enhance immune-mediated tumor control through increased phagocytosis, cell death, and antigen presentation. The NiCOL phase 1 trial (NCT03298893) is designed to determine the safety/tolerance profile and the recommended phase-II dose of nivolumab with and following concurrent CRT in 16 women with locally advanced cervical cancer. Secondary endpoints include objective response rate (ORR), progression free survival (PFS), disease free survival, and immune correlates of response. Three patients experience grade 3 dose-limiting toxicities. The pre-specified endpoints are met, and overall response rate is 93.8% [95%CI: 69.8–99.8%] with a 2-year PFS of 75% [95% CI: 56.5–99.5%]. Compared to patients with progressive disease (PD), progression-free (PF) subjects show a brisker stromal immune infiltrate, higher proximity of tumor-infiltrating CD3$^+$ T cells to PD-L1$^+$ tumor cells and of FOXP3$^+$ T cells to pro-liferating CD11c$^+$ myeloid cells. PF show higher baseline levels of PD-1 and ICOS-L on tumor-infiltrating EMRA CD4$^+$ T cells and tumor-associated macrophages, respectively; PD instead, display enhanced PD-L1 expression on TAMs, higher peripheral frequencies of proliferating Tregs at baseline and higher PD-1 levels at week 6 post-treatment initiation on CD4 and CD8 T cell subsets. Concomitant nivolumab plus definitive CRT is safe and associated with encouraging PFS rates. Further validation in the subset of locally advanced cervical cancer displaying pre-existing, adaptive immune activation is warranted.

Cervical carcinomas are the second leading cause of mortality by cancer in women aged 20 to 39 years, worldwide[1]. Human papilloma-virus (HPV) infection is detectable in nearly 90% of cervical carcinomas and has been recognized as the main causative factor of these tumors, which result from epithelial cell disruption, proliferation, and trans-formation mediated by the E6 and E7 HPV oncoproteins[2]. HPV vaccines against the most frequent serotypes showed great efficacy in pre-venting preneoplastic lesions[3]. However, HPV vaccination coverage is still low in low-income countries and even in some high-income countries such as the United States or France where the current pre-valence among female adolescents is as low as 59%[4]. Non-metastatic, early-stage (i.e., smaller than 4 cm and without detectable nodal

invasion), cervical carcinomas are usually treated by surgery while a combination of chemoradiotherapy followed by brachytherapy has been recommended for the last 24 years for locally-advanced cervical cancer (LACC)[5]. However, LACC is still associated with a 40% risk of disease recurrence[6], which can be significantly improved by optimal image-guided brachytherapy, as reported in the RetroEMBRACE study[7].

The introduction of T cell-targeted immunomodulators blocking the immune checkpoints (ICI) CTLA-4, PD-1, or PD-L1, which aim to harness the host immunity to fight cancer, emerged as a promising strategy in the management of several solid tumors[8]. Cervical cancer is a natural target for ICI as chronic HPV infection induces substantial

✉e-mail: Emanuela.Romano@curie.fr

expression of immune-reactive viral neo-antigens and PD-L1 expression on cervical carcinoma cells[9,10], as shown in pre-clinical studies on cervical carcinoma murine models[11]. The KEYNOTE-826 phase-III trial[12] demonstrated that adding an anti-PD-1 inhibitor, pembrolizumab, to first-line chemotherapy substantially increased median progression-free survival (PFS) from 8.2 months to 10.4 months, and 2-year overall survival (OS) from. 40.4 to 50.4% and led to FDA approval of the combination treatment. Similarly, an interim analysis of the GOG-3016 phase-III trial[13] showed that cemiplimab, a PD-1 inhibitor, improved patient OS when compared to other single-agent chemotherapy in the second metastatic line. Other ICI have also shown compelling results in phase-II trials in cervical cancer such as PD-1 inhibitors: balstilimab[14], nivolumab[15], pembrolizumab[16], camrelizumab[17] and in combination with CTLA-4 inhibitors zalifrelimab[18] or ipilimumab[15].

In the early setting, the CALLA randomized phase-III trial (NCT03830866)[19] evaluating the benefit of adding durvalumab, a PD-L1 inhibitor, to chemoradiotherapy (CRT) in LACC patients did not meet its primary endpoint of PFS improvement with adding durvalumab to standard CRT. To date, immunologic correlates that may inform the selection of responding patients are lacking. In several solid tumors, PD-L1 expression on tumor or stromal cells[20–22] and the presence of tumor-infiltrating lymphocytes (TILs) are associated with benefits to ICI[23]. In this work, we report a favorable safety and tolerance profile of nivolumab in combination with and following CRT. We characterize the immunological features associated with clinical outcomes and show that patients with LACC that display a brisk immune infiltrate in the proximity of PD-L1+ tumor cells, activated, tumor-infiltrating T cells endowed with costimulatory markers, and enrichment in IFN-related pathways benefit from the combination treatment.

## Results

### Patients and treatments

Between November 2017 and July 2020, 21 LACC patients (average age 47,9 (27–77)) were screened for eligibility and 16 patients were included in the study (Table 1 and Fig. 1A). Fourteen patients (87.5%) were diagnosed with squamous cell carcinomas and two patients (12.5%) with adenocarcinomas. HPV was detected in 14 tumors (87.5%); the two HPV-negative tumors were squamous cell carcinomas. All patients received concomitant chemotherapy; two patients (12.5%) received four cycles of cisplatin instead of the five initially planned (due to the occurrence of grade 3 acute kidney failure and of a grade 2 thrombopenia). Fourteen patients (87.5%) received all six nivolumab administrations during the DLT evaluation period; one patient (6.25%) had no nivolumab administration at weeks 9 and 11 for reasons other than DLT, resulting in less than 70% of the planned nivolumab dose during the DLT evaluation period, and was consequently deemed non-evaluable for the DLT assessment (Fig. 1B). All patients received full course radiotherapy and pulse-dose rate brachytherapy. An additional boost to pathological lymph nodes was conducted in 13 patients (81.25%), lombo-aortic irradiation was conducted in seven patients (43.75%). Adjuvant surgery was conducted in four patients (25%); at week 15 for two patients, at week 23 for one patient, and at week 35 for one patient (Fig. 1B).

### Safety

Among the 15 patients evaluable for DLT, three (20%) experienced DLT: two patients had grade 3 hypotension and one developed a grade 3 acute kidney injury, both DLT being considered to be related to cisplatin administration. No death occurred during the DLT evaluation period. Among the 16 patients, five (31.2%) had acute, grade 4 non-DLT adverse events (AE)—all lymphopenias; patients that developed acute, grade 3 non-DLT AE experienced lymphopenia (eight patients, 50%), neutropenia (seven patients, 43.8%), anemia (two patients, 12.5%), hypokalaemia and hypomagnesemia (one patient, 6.25%). All leukopenia events resolved spontaneously without infectious complications (Supplementary Fig. 1B). Following the 11-week DLT assessment period, three additional patients experienced late grade ≥3 AE (12.5%) occurring under maintenance nivolumab, corresponding to one grade 3 immune-related diarrhea, one grade 3 ureteral injury complicated by renal failure in one patient having undergone adjuvant surgery, and one grade 3 lymphopenia. All the treatment-related AE are summarized in Supplementary Table 1. No patient developed a second cancer. The NICOL phase-I trial validated the recommended phase 2 dose of 240 mg q2w nivolumab combined with and following CRT for LACC patients.

### Efficacy

Median follow-up was 23.8 months (range: 3.9–26.2). There was no disease progression during chemoradiotherapy. Two months after brachytherapy completion, the ORR according to the RECIST 1.1 criteria was 93.8% (95% confidence interval (CI): 69.8–99.8): eight patients (50%) had a complete response (CR) and seven patients (43.75%) had a partial response (PR); one patient (6.25%) had disease progression and underwent subsequent surgery. At the last follow-up, a total of 12 patients had responded (11 CR, 1 PR; ORR 75%, 95% CI: [47.6;92.7]; hereafter grouped as "progression-free", PF); while, four patients had progression of the disease (hereafter grouped as PD): three patients while on nivolumab maintenance (at 3, 4, and 5 months after treatment initiation), and one patient after completion of nivolumab maintenance (at eight months after treatment initiation). Of note, two patients with HPV-negative tumors of squamous cell carcinoma histology achieved a complete response to treatment. All

**Table 1 | Patients' characteristics at baseline**

| Characteristics | N | % |
|---|---|---|
|  | 16 |  |
| Age, mean year (range) | 47,9 (27–77) |  |
| BMI, mean kg/m² (range) | 26.6 (19.0–35.4) |  |
| Menopausal status |  |  |
| Yes | 5 | 31.2% |
| No | 11 | 68.8% |
| Performance status |  |  |
| 0 | 8 | 50.0% |
| 1 | 8 | 50.0% |
| Extent of disease (FIGO 2018 staging) |  |  |
| IB3 | 2 | 12.5% |
| IIB | 8 | 50.0% |
| IIIB | 1 | 6.3% |
| IIIC1 | 4 | 25.0% |
| IVA | 1 | 6.3% |
| Histological type |  |  |
| Squamous cell carcinoma | 14 | 87.5% |
| Adenocarcinoma | 2 | 12.5% |
| Differentiation |  |  |
| Poorly differentiated | 4 | 25.0% |
| Moderately differentiated | 6 | 37.5% |
| Well-differentiated | 3 | 18.8% |
| NA | 3 | 18.8% |
| HPV status |  |  |
| Positive | 14 | 87.5% |
| Negative | 2 | 12.5% |
| HPV16 | 6 | 43% |
| HPV18 | 3 | 21% |
| HPV39/45/58/59 | 5 | 36% |

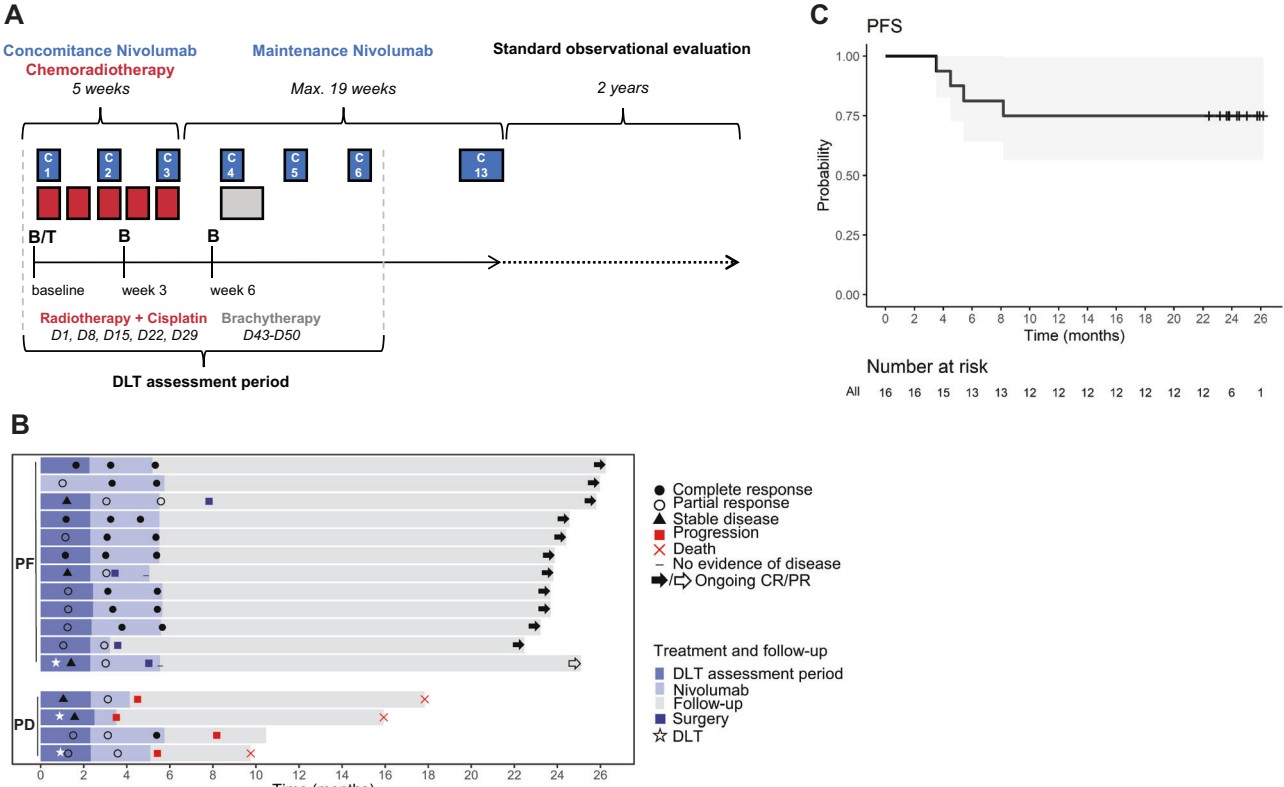

**Fig. 1 | Study design, progression-free survival, and treatment response.**
**A** Study design of the NiCOL Trial, C cycle. B blood sampling, T tumor sampling, D day, DLT dose-limiting toxicity. Red squares symbolize weeks of chemoradiotherapy (five daily fractions of 1.8 G, combined with one intravenous infusion of cisplatin 40 mg/m²); the gray square symbolizes intra-cavitary uterovaginal brachytherapy (85 Gy to the high-risk volume in pulsed dose rate); the blue squares symbolize nivolumab intravenous infusion (240 mg every 2 weeks). **B** Swimmer's plot sorted by treatment response: Progression-free patients (PF, n = 12) in the

upper part, comprise patients with Complete Response (CR, full black arrow and circle, n = 11) and with Partial Response (PR, empty arrow and circle, n = 1); at the bottom, patients with Progression of Disease (PD, n = 4) are highlighted by a red square (all biologically independent samples). A red cross indicates the time of the patient's death. The dose-limiting toxicity (DLT) window, the length of treatment, the time of surgery, and follow-up (in months) are also shown. A white star indicates a DLT event. **C** Progression-free survival (PFS).

patients that progressed had squamous cell carcinoma. Disease progression was loco-regional in three patients and loco-regional plus distant in one patient. Three patients died (18.75%) due to disease progression. Two-year PFS was 75% (95% CI: 56.5−99.5%) (Fig. 1C). Median PFS has not been reached.

## Differential gene sets enrichment in progressors vs non-progressors

To explore the biological differences underlying patient outcome, we inquired whether transcriptomic features would differentiate between PF (n = 11) versus PD (n = 4) samples at baseline (Fig. 2). We first analysed the differentially expressed genes across the two groups but no statistically differentially expressed gene between PF and PD tumors was observed (two-fold difference with a Mann−Whitney uncorrected p value ≤0.1; Fig. 2A). However, gene set variation analysis (GSVA) and gene set enrichment analysis (GSEA) revealed a non-significant enrichment of epithelial-to-mesenchymal transition (EMT; GSEA adjusted p value [adjPval] = 2,25e-21; GSVA fold change [FC] = 2.25), angiogenesis (GSEA-adjPval = 1.73e-2; GSVA-FC = 2.02) and KRAS signaling (GSEA-adjPval = 2.21e-4; GSVA-FC = 1.57) gene sets in PD; whereas, PF were enriched in interferon-(IFN-) alpha (GSEA-adjPval = 2.19e-5; GSVA-FC = 0.67) and IFN-gamma (GSEA-adjPval = 5.9e-3; GSVA-FC = 0.78 Fig. 2B−D and Supplementary Data 1). At the DNA level, using a dedicated NGS gene panel, the main observed alterations across all patients were in FAT1 (30%), STK11 (30%), CASP8 (20%), PIK3CA (20%), and YAP1 (20%). No KRAS, NRAS, or HRAS mutation was

found. Two tumors were TMB high, including one PF and one PD (Supplementary Fig. 1C).

## Brisker, preexisting, intratumor immune infiltrate at baseline in responder patients

According to the clinical outcome, we evaluated the presence of tumor-infiltrating immune cells in progression-free patients (PF) versus (vs) patients with progressive disease (PD) in the tumor nests and in the tumor-surrounding stroma (stroma). At baseline, there was no difference in the abundance of tumor-infiltrating leukocytes (TILs) by hematoxylin and eosin (H&E) in PF vs PD (Fig. 3A and Supplementary Fig. 2A). We then assessed by multiplexed immunohistochemistry (mIHC) the presence of tumor-infiltrating CD3⁺ (total T cells), Granzyme B⁺ (GZMB⁺, a proxy marker of cytotoxic T cells), FOXP3⁺ (marker of CD4⁺ regulatory T cells, Tregs), CD11c⁺ (myeloid cells), and the expression of PD-L1, by both immune/stromal and tumor cells of PF and PD (Fig. 3B−D and Supplementary Fig. 2B, C). Interestingly, the tumors of PF patients showed a significant increase in stromal CD3⁺, GZMB⁺, FOXP3⁺, and CD11c⁺ cells compared with the tumor area (Fig. 3C). Non-tumoral PD-L1⁺ cells−comprising immune and stromal cells−significantly accumulated in the stroma of PF patients, along with an increase, although not significant, of PD-L1⁺ tumor cells (expressing cytokeratins, CKs) in the tumor areas of PF patients (Fig. 3D and Supplementary Fig. 2C). Moreover, in PF patients, we observed a positive correlation between the intratumoral density of CD11c- and CD3-expressing cells (Supplementary Fig. 2D). Thus, these

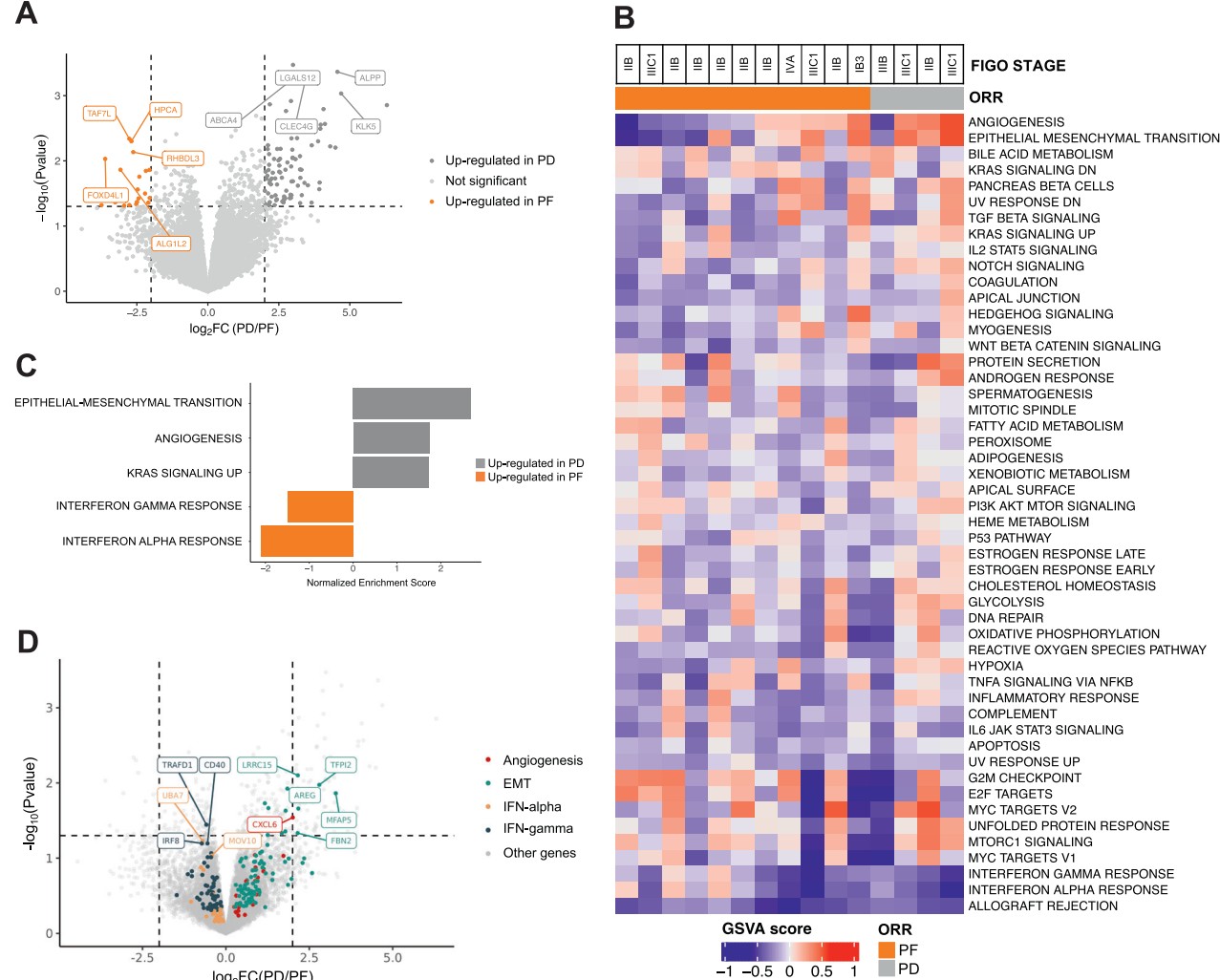

**Fig. 2 | Bulk RNA sequencing and genomic alteration assessment of baseline tumors. A** Volcano plot showing log2-transformed Fold Change (log2FC) in protein-coding genes in patients with progression of disease (PD) versus progression-free ones (PF). While no statistically significant gene was detected, orange and dark gray dots correspond to the most upregulated genes in PF and in PD samples, respectively. **B** Gene set variation analysis (GSVA) on MSigDB Hallmark gene sets. PF and PD samples are annotated in orange and gray, respectively. **C** Gene set enrichment analysis (GSEA) showing epithelial-mesenchymal transition (EMT), angiogenesis, *KRAS* signaling up, interferon-gamma response, and interferon-alpha response Hallmark gene sets from MSigDB that are significantly enriched in the tumors of PF

(orange) versus PD (gray) patients (Benjamini–Hochberg adjusted $p < 0.05$). Fold change expression values and adjusted $p$ values were combined to rank the genes as input for the GSEA which is based on the Kolmogorov–Smirnov test. **D** Volcano plot highlighting the protein-coding genes with the highest FC for the following signatures: Angiogenesis (in red) and EMT (in green) pathways—enriched in PD; Interferon-alpha (yellow) and interferon-gamma (dark blue) pathways—enriched in PF. None of the highlighted genes was statistically significant. In all panels, $n = 11$ for PF and $n = 4$ for PD (biologically independent samples). The complete list of the genes that are enriched in the pathways of the GSEA analysis are provided in Supplementary Data 1.

data suggest that PF patients have a preexisting higher immune activity compared with PD patients.

Next, we quantified the proliferating bona fide cytotoxic CD3 (GZMB$^+$), FOXP3$^+$, CD11c$^+$ immune cells, and tumor cells (Pan-CK$^+$) in the stroma and tumor areas by staining for the proliferation marker Ki67 (Fig. 3E, F and Supplementary Fig. 2E). Of note, the stroma of PF patients harbored significantly higher numbers of proliferating cytotoxic GZMB$^+$ and Treg cells compared with the tumor area (Fig. 3F).

To address whether the preexisting immune-reactive landscape observed in PF patients is endowed with antigen-presenting cells (APCs) to efficiently prime T cells, we explored the expression of CD28 and PD-1 on CD8$^+$ T cells and CD86 on the CD11c$^+$ compartment (Fig. 3G, H and Supplementary Fig. 2F). PF and PD displayed similar numbers of CD11c$^+$ CD86$^+$ APCs in the tumor and stroma areas (Fig. 3H). PF displayed significantly higher numbers of CD8$^+$ CD28$^+$ (PD-1$^-$) cells in the stroma compared with the tumor (Fig. 3H and Supplementary Fig. 2F). Overall, these data suggest that PF patients have a

more reactive TME, characterized by higher numbers of CD28-expressing T cells before treatment.

As the proximity of T cells to tumor cells has been linked with response to ICIs[24], we interrogated the distance between T cells and tumor cells and between T cells and APCs (Fig. 3I–L), thanks to the nearest neighbor analysis module[25], which measures the average distance between two cell types. Indeed, we observed that in the stroma of PF patients, CD3$^+$ cells were significantly closer to PD-L1$^+$ tumor cells as opposed to PD patients (Fig. 3I, J). In addition, we observed that FOXP3$^+$Ki67$^+$ cells were closer to proliferating CD11c$^+$ cells in the stroma of PF patients, compared with PD (Fig. 3K). Finally, although not significant, in progression-free subjects—CD8$^+$ CD28$^+$ PD-1$^+$ T cells were in closer proximity to both APCs (CD11c$^+$ CD86$^+$) and tumor cells (CK$^+$) especially in the tumor-surrounding stroma, compared to PD (Fig. 3L).

Taken together, these data demonstrate that, unlike PD, PF patients show higher proximity of tumor-infiltrating T cells to tumor cells, and of proliferating Tregs to CD11c$^+$ cells in the stroma.

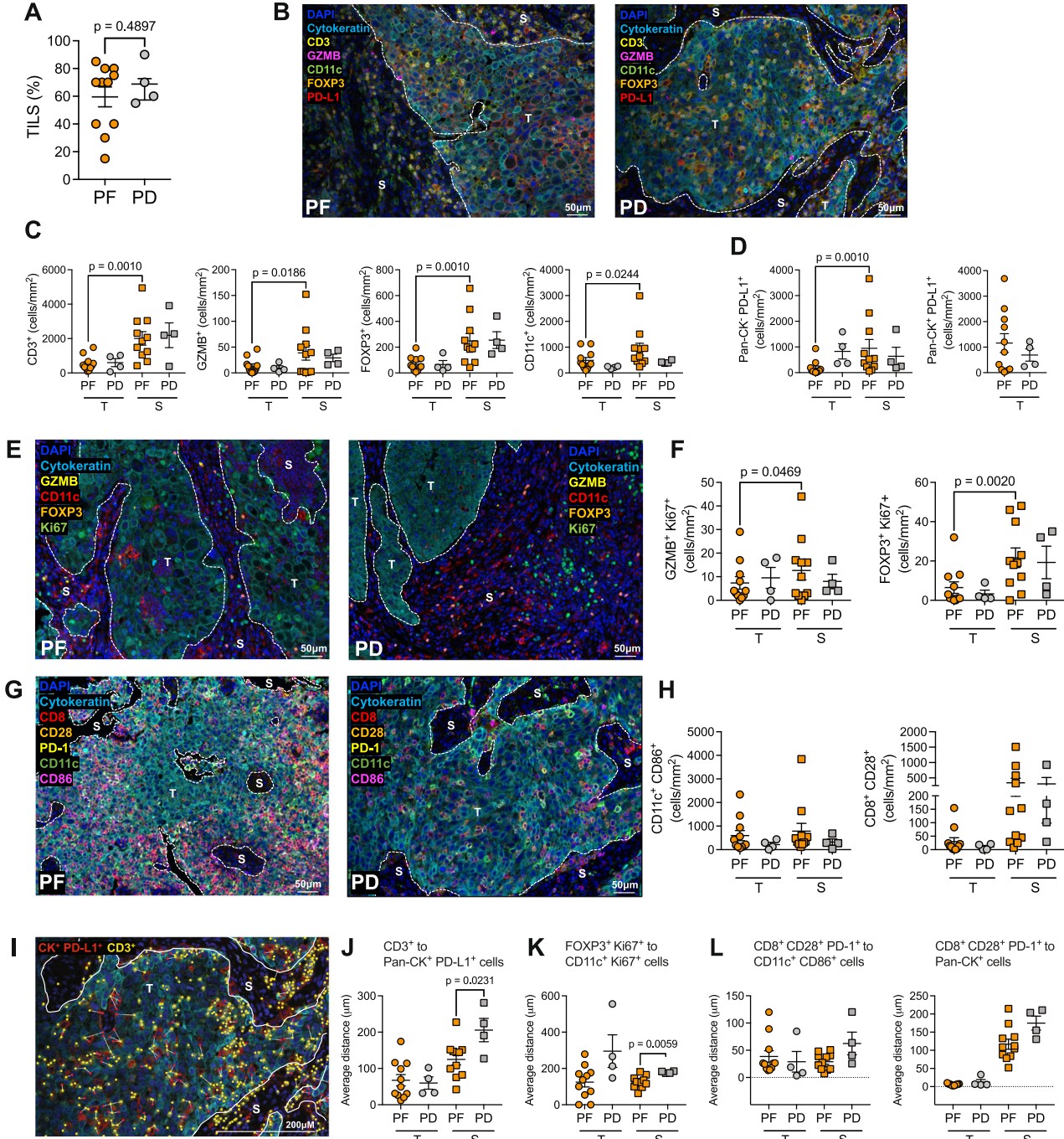

**Fig. 3 | Profiling of intratumoral T and myeloid cells. A** Percentage of TILs, assessed by H&E staining, in PF vs PD. **B** Representative Multiplex Immunohistochemistry (mIHC) staining of the tumor microenvironment (TME) for CD3+, Granzyme B (GZMB+), FOXP3+, CD11c+, Pan-cytokeratin (Pan-CK) tumor cells and PD-L1+ cells in PF vs PD. **C** Number of CD3+, GZMB+, FOXP3+, CD11c+ cells/mm² in the S vs T areas in PF *vs* PD. **D** Number of PD-L1+ Pan-CK- cells/mm² (i.e., non-tumor cells) in the S vs T areas (left), and of PD-L1+ Pan-CK+ cells in the T area in PF vs PD. **E** Representative mIHC staining of the TME for GZMB+, FOXP3+, CD11c+, Pan-CK+ tumor cells, and Ki67+ cells in PF vs PD. **F** Number of proliferating (Ki67+) GZMB+ (left) and FOXP3+ (right) cells/mm² in the S vs T areas, in PF vs PD. **G** Representative mIHC staining of the TME for CD8+, CD28+, PD-1+, CD11c+, CD86+, and Pan-CK+ tumor cells in PF vs PD. **H** Number of CD11c+CD86+ double positive antigen-presenting cells/mm² (APC) (left) and CD8+CD28+ cells/mm² (right) in the S vs T areas, in PF vs

PD. **I** Representative mIHC staining (from a PD) highlighting CD3+ cells (yellow dots) and PD-L1+ Pan-CK+ tumor cells (red dots). The white lines represent the intercellular distance between each CD3+ and PD-L1+ tumor cell in proximity. **J** Cumulative data of the proximity of CD3+ cells to PD-L1+ tumor cells in the S and T areas of PF vs PD. **K** Cumulative data of the proximity of FOXP3+ Ki67+ cells to CD11c+ Ki67+ cells in the S area of PF vs PD. **L** Cumulative data of the proximity of CD8+ CD28+ cells to CD11c+ CD86+ APC (left) and to Pan-CK+ cells (right) in the S area of PF vs PD. In all panels, $n = 11$ for PF and $n = 4$ for PD (biologically independent samples). Panels **B**, **E**, **G**, and **I** are representative of 15 biologically independent samples. Data were presented as individual values (mean ± SEM). Statistical tests: two-tailed unpaired and paired *t*-tests (Mann–Whitney and Wilcoxon matched-pairs signed rank test) in all panels. T tumor, S Stroma. Source data are provided as Source Data File.

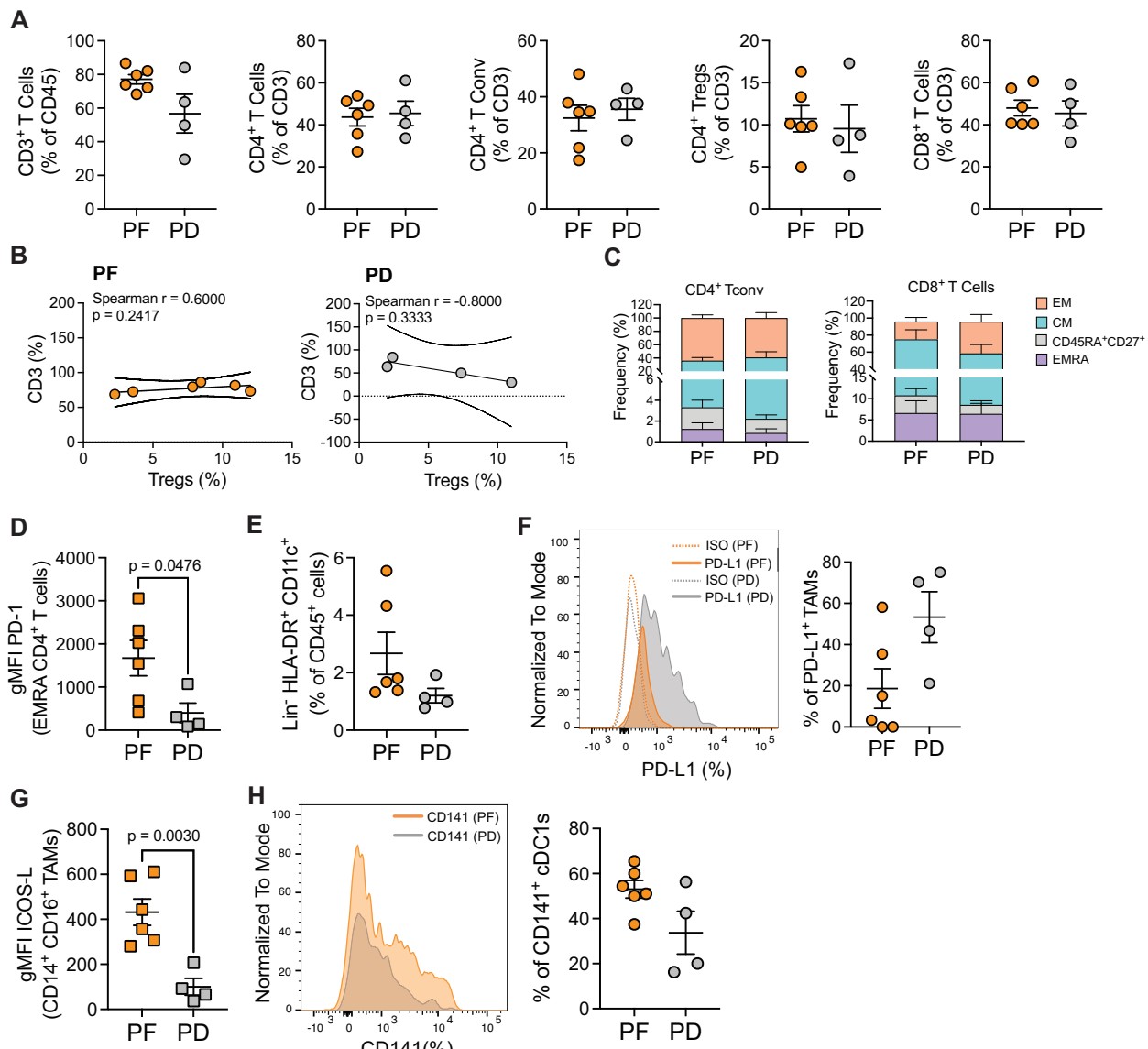

**Fig. 4 | Ex vivo phenotyping of intratumoral T and myeloid cell subsets by flow cytometry. A** Frequency of intratumoral CD3⁺ T cells gated on CD45⁺ cells, CD4⁺ T cells, CD4⁺ Tconv (CD3⁺CD4⁺CD127⁻/⁺CD25⁻), CD4⁺ Tregs (CD3⁺CD4⁺CD127ˡᵒCD25⁺/ʰⁱᵍʰ) and of CD8⁺ T cells, expressed as a percentage of CD3⁺ T cells, in PF vs PD. **B** Spearman correlation between the intratumoral frequency of CD3 and Tregs (both expressed as a percentage of CD45⁺ cells), assessed by FC in PF vs PD. The graphs display non-linear regression curves and 95% confidence intervals. **C** Distribution of CD4⁺ and CD8⁺ T cell subsets (EMRA = CD45RA⁺CD27⁻; CD45RA⁺CD27⁺ cells; CM = CD45RA⁻CD27⁺; EM = CD45RA⁻CD27⁻) expressed as a frequency of CD4⁺ and CD8⁺ T cells, respectively, in PF vs PD. **D** Mean fluorescence intensity (MFI), expressed as Geomean (gMFI) of PD-1 in EMRA CD4⁺ T cell compartment in PF vs PD. **E** Distribution of the total myeloid compartment, defined as Lineage⁻ (CD3⁻CD19⁻CD56⁻), HLA-DR⁺CD11c⁺, in PF vs PD. **F** Representative histograms showing the frequency of PD-L1⁺ TAMs (gated on Lineage⁻HLA-DR⁺CD11c⁺CD14⁺CD16⁺CD163⁺CD206⁺) in PF vs PD (plain color in orange and gray, respectively) and isotype controls (relative dotted lines) are shown on the left. Cumulative data are shown on the right. **G** gMFI of ICOS-L in Lineage⁻HLA-DR⁺CD11c⁺CD14⁺CD16⁺CD163⁻CD206⁻ TAMs. **H** Frequency of conventional DC1 (cDC1) cells, expressed as a frequency of CD141⁺ cells, within lineage⁻HLA-DR⁺CD11c⁺CD14⁻CD16⁻ cells, in PF vs PD. In all panels, PF: *n* = 6; PD: *n* = 4 (biologically independent samples). Data were presented as individual values showing mean ± SEM. Statistical tests: two-tailed unpaired *t*-test (Mann–Whitney test) (Panels **A**, **C**–**H**). Nonparametric Spearman correlation (Panel **B**). If not indicated, no statistically significant difference was observed. Tconv T conventional, Treg T regulatory, EM effector memory, CM central memory, EMRA effector memory RA. Source data relative to all panels are provided as Source Data File.

## More activated, preexisting immune cells in the tumor of progression-free patients

To assess the differences in the tumor-infiltrating immune cell compartments between PF and PD, a deeper characterization by multi-parametric FC was performed. In line with the TIL assessment, we confirmed similar frequencies of tumor-infiltrating T cells, such as total CD3⁺ cells, conventional CD4⁺ T conventional (Tconv) cells, CD4⁺ Tregs and CD8⁺ T cells in PF and PD patients (Fig. 4A and Supplementary Fig. 3A). PF patients showed a non-significant increase in CD3⁺ T cells frequencies (Fig. 4A).

A positive correlation between the frequency of CD3⁺ T cells and CD4⁺ Tregs was observed in PF, whereas PD tended to have a negative correlation, although not significant (Fig. 4B). These results suggest that, despite quantitatively similar immune infiltrates, PF patients could experience lower Treg-driven inhibition in the tumor before starting the treatment compared with PD.

We then inquired the frequencies of both CD4⁺ and CD8⁺ T cell subsets, including effector memory RA (EMRA), CD45RA⁺CD27⁺ cells, central memory (CM), and effector memory (EM) cells. No significant differences in their relative distribution were observed between PF and

PD (Fig. 4C). Although the frequency of PD-1⁺ cells was comparable both in total CD8⁺ and CD4⁺ Tconv and Treg cells (Supplementary Fig. 3B), as well as in CD4⁺ and CD8⁺ T cell subsets (Supplementary Fig. 3C), tumor-infiltrating terminally differentiated EMRA CD4⁺ T cells were among the subsets with significantly higher PD-1 expression, at the mean fluorescence intensity (MFI) level, in PF vs PD (Fig. 4D). Globally, these data show that PF patients have a positive CD3-to-Treg correlation and higher, preexisting levels of PD-1 expression in EMRA CD4⁺ T cell subset, compared with PD.

Next, we assessed the frequency and the phenotype of the myeloid subpopulations in PF and PD patients (Supplementary Fig. 3D). Interestingly, the global myeloid compartment (defined as Lineage⁻ (CD3⁻CD19⁻CD56⁻) HLA-DR⁺ CD11c⁺) infiltrated similarly the tumors of both PF and PD (Fig. 4E). To characterize the phenotype of TAMs, we evaluated the expression of the costimulatory/inhibitory markers PD-L1 and ICOS-L by FC. PD patients displayed higher frequencies of PD-L1⁺ TAMs (co-expressing CD14, CD16, CD163, and CD206) (Fig. 4F). No significant differences were highlighted in TAM subpopulation frequencies, according to the expression of CD14, CD16, and the "M2" markers CD163 and CD206 (Supplementary Fig. 3E–G). PF patients had higher expression of ICOS-L on CD14⁺CD16⁺ TAMs (Fig. 4G). Of note, we observed a non-significant increase in the proportions of CD141⁺ cDC1 cells in PF (Fig. 4H), which could suggest an enhanced antigen presentation taking place in the tumor of PF patients. Taken together, these data suggest that the clinical response might depend on the preexisting host immunity, being more active in PF patients.

### Baseline accumulation of highly proliferating Tregs and enhanced PD-1 expression in CD4⁺ and CD8⁺ T cell subsets on treatment in the blood of PD patients

Blood samples were collected at baseline, week 3 and week 6 on treatment. After PBMC isolation, the T and myeloid cells were analysed by multiparametric FC (Fig. 5 and Supplementary Fig. 4A, J). Although there were no differences in CD4⁺ Tconv frequencies at baseline and week 3 in PF and PD, we observed a higher frequency of CD4⁺ Tconv in PF compared with PD at week 6—a later on-treatment time point (Fig. 5A), suggesting persisting CD4⁺ T cell responses in PF patients over time. Interestingly, despite the lower numbers of available samples at week 6, the CD4⁺ Tconv cells of PD had significantly higher gMFI levels of PD-1 and contained higher frequencies of PD-1⁺ cells, compared with PF (Fig. 5B, C). No differences in Tregs and CD8⁺ T cells were observed among the three time points in PF vs PD patients (Fig. 5A). In addition, the frequency of PD-1⁺ cells were comparable in the Treg and CD8⁺ T cell compartments across the different time points, in both PF and PD cohorts (Supplementary Fig. 4B, C). Next, we analysed the percentages of circulating CD4⁺ Tconv cell subsets according to the expression of CD45RA and CD27 markers. The frequency of EMRA, naïve, CM, and EM CD4⁺ T cells resulted to be comparable between PF and PD across the three time points (Supplementary Fig. 4D). Of note, the EMRA and EM CD4⁺ T cells expressed significantly more PD-1 at week 6 in PD patients (Fig. 5D, E), whereas the naïve and CM subsets showed similar levels of PD-1 expression in the PF and PD across the three time points Supplementary Fig. 4E).

Finally, EMRA CD4⁺ T cells, expressed higher levels of OX40 at week 6 in PF patients, compared to PD, and this level was comparable, within PF, across the three time points (Fig. 5F). Interestingly, we observed that PD subjects had significantly more Ki67⁺ Tregs at baseline compared to PF ones (Fig. 5G), a difference that was lost at later time points, suggesting a preexisting Treg proliferation in PD patients. No differences in the proliferative capacity in CD4⁺ Tconv and CD8⁺ T cells was observed in PF vs PD at the different time points (Supplementary Fig. 4F, I).

Despite similar proportions of CD8⁺ T cell subsets (Supplementary Fig. 4G) and similarly to EM CD4⁺ Tconv (Fig. 5E), we detected a higher fraction of EM CD8⁺ T cells expressing PD-1 in PD patients at

week 6 (Fig. 5H), suggesting a global shift to higher PD-1 expression in EM T cells of PD patients, that could control the T cell effector function on treatment. No differences in PD1⁺ EMRA, naïve, and CM CD8⁺ T cell subsets were observed between PF and PD across the three time points (Supplementary Fig. 4H).

In the myeloid compartment, similar proportions and phenotypes of CD14⁺ classical, CD14⁺ CD16⁺ intermediate and CD16⁺ non-classical monocytes (Supplementary Fig. 4J, K), as well as of CD141⁺ cDC1s, CD1c⁺ cDC2s, and CD123⁺ pDCs (Supplementary Fig. 4J, L) were observed, in both PF and PD groups, indicating that myeloid cells are similarly represented in both PF and PD and that neither their frequencies nor their phenotypes are modulated upon treatment in blood.

Taken together, our data show that, in PD patients, the peripheral CD4⁺ and CD8⁺ T cell subsets increase PD-1 expression only at later time points, possibly correlating with a less functional phenotype. Moreover, the higher proportion of circulating Ki67⁺ proliferating Tregs at baseline might indicate a preexisting immunosuppressive environment, that may lead to inefficient antitumor immune responses.

## Discussion

The NICOL phase-I trial validated the recommended phase 2 dose of 240 mg q2w nivolumab combined with and following CRT for LACC patients. Indeed, although three DLT were observed, these toxicities corresponded to hypotension and acute kidney failure, which were unrelated to nivolumab but rather to cisplatin administration. The acute toxicity profile of this triple combination was similar to what is commonly observed with cisplatin CRT for LACC[26]. With a median follow-up of 23.8 months, only one patient experienced an immune-related AE, corresponding to a manageable grade 3 diarrhea. We observed a 25% rate of acute, grade 4 lymphopenia, likely related to radiotherapy, especially when associated with cisplatin. Lymphopenia has been reported during CRT[27,28] and it should be noted that it was transitory and not associated with opportunistic infections. The incidence of severe lymphopenia was, however, much more pronounced than in the REACH and JAVELIN Head and Neck 100 trials both evaluating avelumab and CRT (with cetuximab and cisplatin, respectively) in locally-advanced squamous cell carcinoma of the head and neck, where the rate of grade 4 lymphopenia ranged between 2 and 3%[29,30]. Interestingly, the onset of lymphopenia might be an early indicator of the risk of immune-related adverse events[31]. Furthermore, baseline and persistent lymphopenia while under ICI treatment is predictive of a shorter time to progression[32]. These observations should justify repeated laboratory tests. Concomitant and maintenance nivolumab at full dose (i.e., 240 mg q2w) can be reasonably recommended for future trials assessing the efficacy of this treatment combination.

In agreement with previously reported data in advanced melanoma[33] and in spite of the small sample size preventing to detect substantial differences at the transcriptomic level, we observed that PF tumors were enriched at baseline for gene sets linked to response to ICI, i.e., IFN-related pathways in multiple solid tumors[33–35]; whereas, PD patients were enriched in EMT, angiogenesis-related and KRAS gene sets, previously associated with resistance to ICI[36–39]. FAT1, STK11, CASP8, PIK3CA, and YAP1 were previously described in cervical cancer[40–42] and were among the most recurrent alterations found in this study cohort. These genes encode for proteins known to be involved in cell proliferation and metastasis inhibition, regulation of cell polarity and metabolism, regulation of apoptosis, oncogenic transformation, and therapy resistance[40,42–45]. However, possibly due to the limited number of patients, we could not find any correlation between the presence of one of these alterations and ORR.

The synergy between angiogenesis targeting and immune checkpoint inhibition is under investigation in numerous cancer types[46]. Of note, the KEYNOTE-826 phase-III clinical trial evaluated the association of chemotherapy with or without bevacizumab and

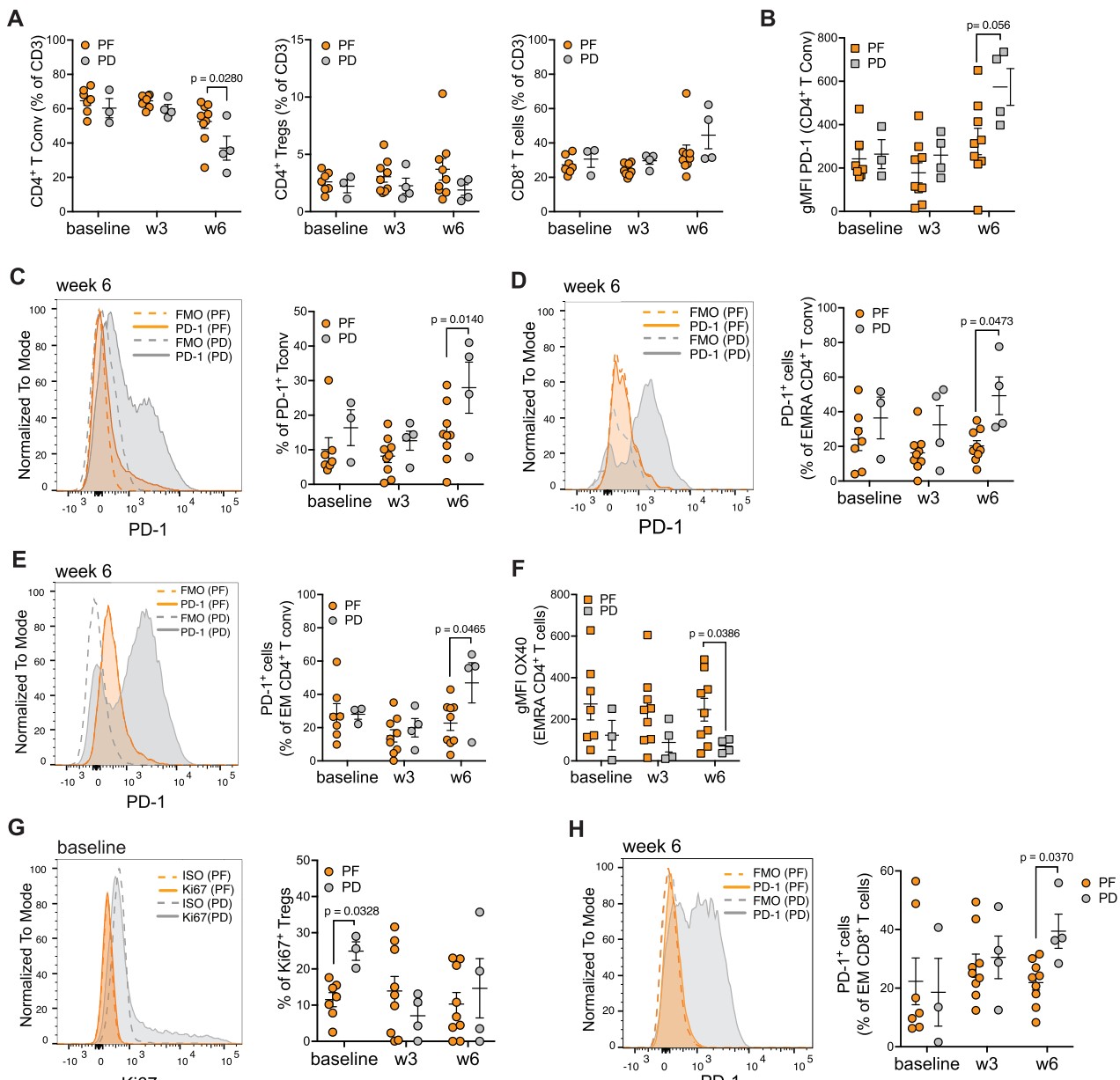

**Fig. 5 | Immune monitoring of peripheral T cells by flow cytometry. A** Frequency of peripheral CD4+ Tconv (CD3+CD4+CD127+/−CD25−), CD4+ Tregs (CD3+CD4+CD127loCD25+/high), and CD8+ T cells in PF vs PD at the indicated time points. **B** gMFI of PD-1 in CD4+ T conv cells in PF vs PD at the indicated time points. **C** Representative histograms at week 6 (left) and cumulative data (right) of the frequency of PD-1+ CD4+ Tconv cells in PF vs PD (in orange and gray, respectively), with isotype controls (relative dotted lines) at the indicated time points. **D** Representative histograms at week 6 (left) and cumulative data (right) of the frequency of PD-1+ EMRA CD4+ T cells in PF vs PD (in orange and gray, respectively), isotype controls (relative dotted lines) at the indicated time points. **E** Representative histograms at week 6 (left) and cumulative data (right) of the frequency of PD-1+ EM CD4+ Tconv cells in PF vs PD (in orange and gray, respectively), with isotype controls (relative dotted lines) at the indicated time points.

**F** gMFI of OX40 in EMRA CD4+ T cell subset in PF vs PD at the indicated time points. **G** Representative histograms at baseline (left) and cumulative data (right) of the frequency of Ki67+ Tregs in PF vs PD (in orange and gray, respectively), isotype controls (relative dotted lines) at the indicated time points. **H** Representative histogram showing the frequency of PD-1+ EM CD8+ T cells at week 6 in PF vs PD (in orange and gray, respectively), isotype controls (relative dotted lines) (left). Cumulative data were shown on the right. In all panels: n = 7 for PF, n = 3 for PD at baseline; n = 9 for PF, n = 4 for PD week 3 and week 6 (biologically independent samples). Data were presented as individual values showing mean ± SEM. Statistical test: Two-way ANOVA−mixed-effects model with the Geisser-Greenhouse correction (all panels). If not indicated, no statistically significant difference was observed. Tconv T conventional, Treg T regulatory, EM effector memory, CM central memory, EMRA effector memory RA. Source data are provided as Source Data File.

pembrolizumab/placebo in persistent, recurrent, or metastatic cervical cancer[12]. In the subgroup analysis, the addition of bevacizumab showed a trend of improved efficacy (Hazard Ratio [HR] = 0.63 [0.47−0.87] versus 0.74 [0.53−1.04] for patients without bevacizumab), suggesting that adjuvant anti-angiogenic therapy combined with immune checkpoint inhibition warrants further investigation.

The CALLA study−a phase-III, randomized trial (NCT03830866)[19] evaluating the benefit of adding durvalumab, a PD-L1 inhibitor, to CRT in LACC patients−did not meet its primary endpoint. The two-year PFS and ORR in the NiCOL study are comparable and within the HR of the CALLA study, although we should take into account the shorter follow-up and the limited number of patients in the NiCOL trial. Furthermore, it should be emphasized that the immunologic targets and the

intracellular signaling of PD-L1 vs PD-1 inhibitors are different[47,48] and that their tolerance profile overlaps only partially[49,50].

Of note, nor tumor histology or HPV status were associated with an outcome of the combination of CRT and PD-1 inhibition in our study. In addition, PD-L1 expression on tumor cells per se did not enrich for response to PD-1 blockade, unlike it was reported in studies testing the efficacy of pembrolizumab in metastatic or unresectable cervical cancer[12,16]. Such discrepancies in the robustness of PD-L1 status to predict the efficacy to anti-PD-L(1) therapy in the early vs metastatic cancer setting have been also reported in triple-negative breast cancer[51,52]. Large, randomized trials are still required to evaluate the benefit of PD-1 inhibitor combination with CRT in squamous cell carcinomas. The ongoing KEYNOTE-A18 randomized phase-III trial[53] is evaluating pembrolizumab in combination with CRT and in maintenance for LACC; results are expected in 2024. In addition, nivolumab in combination with ipilimumab before CRT and as maintenance in LACC patients has recently been evaluated in the COLIBRI trial and results are awaited[54]. For optimal clinical trial design and interpretation of current studies of immunotherapy in LACC patients, it is important to note the potentially detrimental effect of nodal irradiation on the antitumor immune response[55]. Nodal irradiation decreases epitope spreading and adaptive immune responses through altered chemokine expression and CD8[+] T-cell trafficking[56]. According to the recent literature, priming of tumor-specific CD8[+] T cells occurs in tumor-draining lymph nodes with the subsequent acquisition of effector capacity within the tumor after effective co-stimulation[57]. Similarly, nodal irradiation might also partially explain the deceiving results of the REACH and JAVELIN Head and Neck 100 trials in patients with locally-advanced head and neck, squamous cell carcinomas, although immune checkpoint inhibition provides PFS and OS benefit in patients with such carcinomas in the[30,58]. These studies can provide important lessons for the future, as we should perhaps restrict lymph node irradiation in protocols associating CRT with immunotherapy.

Regarding the immunological correlates of clinical outcome, and in line with the transcriptomic data, we observed that PF patients display higher numbers of proliferating, Ki67[+] cytotoxic GZMB[+], FOXP3[+] regulatory T cells in the stroma compared to the tumor, along with higher PD-L1[+] expression on immune cells in the stroma. Also, higher numbers of CD8[+]CD28[+] T cells were found in PF, especially in the stroma. Progression-free patients have a globally higher preexisting immune cell infiltrate compared with PD patients. Indeed, although not significant, the intratumoral immune landscape of PD appears to be weaker, in both S and T areas, compared with PF patients. Moreover, we observed a positive correlation between intratumoral CD3[+] T cells and total CD11c[+] myeloid cells in PF patients suggesting a preserved T cells/myeloid infiltrate. Unlike PD, progression-free patients show PD-L1 positive tumor cells in closer proximity with CD3[+] T cells. Proximity of immune and tumor cells has been associated with response to anti-PD-1-based therapies in metastatic melanoma[24]. Interestingly, in the stroma, CD8[+]CD28[+]PD-1[+] T cells were also found to be closer to both tumor cells and to CD11c[+]CD86[+] APCs in progression-free patients. These results are in line with pre-clinical models[59,60] indicating that CD28 co-stimulation is associated with effective PD1-directed therapy and suggest that CD28 expression on TILs may serve as a potential biomarker to predict responsiveness to treatment.

With respect to intratumoral Tregs, although they have been often associated with poorer clinical outcome[61], including in ovarian carcinoma[62], in the NiCOL study we report a higher number of Tregs in PF patients, which were in closer proximity of proliferating CD11c[+] myeloid cells in S areas of PF patients. Our data suggest that Tregs could support myeloid cell functions[63] and their abundance in PF tumors might reflect the control of an active antitumor immune response, that is not present in PD patients. Altogether, these data support a positive correlation between the clinical outcome and the antitumor immune microenvironment in PF.

In line with the clinical outcome and with the immune infiltrate features, PD—unlike PF patients—displayed enhanced PD-1 expression in both peripheral CD4[+] and CD8[+] T cells but only at later time points on treatment, which could indicate a more exhausted state of circulating T cells[64]. Moreover, at baseline, PD patients showed a significantly higher frequency of circulating, proliferating Tregs, compared with PF patients, which instead displayed frequencies similar to healthy individuals[65]. We speculate that PD patients might experience a more pronounced, systemic Treg-mediated immune suppression.

NiCOL is the first clinical trial to report on immune correlates of response to nivolumab with and following CRT in patients with LACC. Its limits are inherent to those of phase-I trials, in particular the limited number of patients, the absence of a comparator arm, the short follow-up, and the lack of paired pre- and on-treatment tumor biopsies. In particular, given the absence of a comparator arm, the biomarker data need to be interpreted with caution. Aware that further testing in randomized clinical trials is warranted, we propose that nivolumab with and following CRT may prove beneficial in LACC patients, whose tumors display a brisk, pretreatment immune infiltrate in the proximity of PD-L1[+] tumor cells, activated, tumor-infiltrating T and myeloid cells along with enrichment in IFN-related pathways.

## Methods

### Study population

The NiCOL trial (NCT03298893) is an open-label, single-arm, phase-I, dose-confirmation, multicenter trial aiming to determine safety and tolerance, and immune correlates of concurrent and maintenance nivolumab plus CRT in LACC patients. Inclusion criteria included immunotherapy-naïve adult patients with histologically confirmed cervical adenocarcinoma or cervical squamous Federation of Gynecology and Obstetrics (FIGO) 2018 stages IB3-IVA, with an indication for curative intent cisplatin-based CRT. The disease had to be amenable to pretreatment biopsy. On-treatment biopsy was not allowed in the study. Exclusion criteria included distant metastatic disease, prior history of radiotherapy, systemic antineoplastic treatment, or clinically significant comorbidities. The Clermont-Ferrand ethic committee (CPP Sud-Est VI, AU 1316) approved the trial, which was conducted in accordance with the Declaration of Helsinki and the national regulatory requirements. All patients signed a written informed consent. The first patient was enrolled on 11/27/2017 and the last patient on 01/20/2020. An overview of the type of translational analyses performed and the number of datasets collected is shown in Supplementary Fig. 1A.

### Study design, endpoints, and statistical analysis

The primary endpoint was the incidence of dose-limiting toxicities (DLT) within 11 weeks after the initiation of treatment, corresponding to the first six cycles of nivolumab. DLT were defined as grade ≥3 non-hematological toxicities, grade ≥3 immune-related adverse events, persistent grade ≥2 immune-related adverse events for more than 1 week despite optimal supportive care, or a radiotherapy delay greater than 1 week related to treatment toxicity. DLT were graded according to the Common Terminology Criteria for Adverse Events (CTCAE), version 4.03. Secondary endpoints included overall response rate (ORR), PFS, and treatment tolerance profile. The overall response rate was radiologically defined according to RECIST 1.1 criteria based on thoracic-abdominal-pelvic computed tomography, completed with pelvic MRI and 18F-FDG PET-CT evaluations. Exploratory analyses included immunological and molecular correlates of response to therapy.

The study was based on a 3 + 3 design in order to confirm the safety of a flat dose of nivolumab at 240 mg q2wk; a maximum of one DLT for six patients was considered acceptable. A single reduced dose level (1-mg/kg) was planned. An expansion cohort of nine

additional patients treated at the determined safe dose level was additionally planned. Consequently, the minimum sample size was 15 evaluable patients, and the maximum sample size was 21 evaluable patients. Patients who failed to complete the 11-week DLT evaluation period or who received less than 70% of the planned dose of nivolumab, of chemotherapy, or of radiotherapy for a reason other than DLT would be considered as not evaluable for the DLT assessment. A Data Safety Monitoring Board was consulted to review the safety profile of the treatment in the first six DLT evaluable patients and in order to proceed with the expansion cohort. The objective response rate (ORR) was defined as the proportion of all subjects whose best response was either a complete response (CR) or a partial response (PR), as assessed using RECIST. The PFS was defined as the duration from the start of the treatment to disease progression, date of last follow-up, or death, regardless of the cause of death. PFS was estimated by the Kaplan–Meier method. The survival analyses were performed using R v4.1.2 software.

GraphPad Prism v9 software was used to perform the statistical analyses of the immunological correlates and the test used is indicated in the legend of each figure. Values were expressed as mean ± SEM or median of biological replicates, as specified. Without mention, differences are not statistically significant. Correlations were calculated using the nonparametric Spearman's correlation test, two-tailed. The study protocol is available as Supplementary Note in the Supplementary Information File.

## Procedures

Nivolumab was administered intravenously at the initial flat dose level of 240 mg q2w with concurrent CRT and maintained for 6 months after CRT completion, corresponding to a total of 13 cycles. Cisplatin was administered intravenously at 40 mg/m² every week, starting on day one of radiotherapy for a total of five cycles. Patients were treated with intensity-modulated radiation therapy technique, using linear accelerators. A dose of 45 gray (Gy) in 25 fractions was delivered to the gross tumor volume with a 2-cm margin, the vagina, the uterine cervix and corpus, the obturator nodes, the internal-external iliac nodes, the common iliac nodes, and the presacral nodes. The para-aortic area and common iliac nodes could also be treated in case of tumor involvement. Simultaneous integrated boost could be delivered to pathological lymph nodes to the total dose of 54 to 57.5 Gy in 25 fractions. Image-guided pulse-dose-rate brachytherapy was delivered on week 7 or 8; brachytherapy target volumes were determined according to the GYN GEC-ESTRO guidelines (13) to ensure homogeneity and reproducibility between the different centers. A total dose of 85 Gy was planned to be delivered to 90% of the high-risk volume and a dose of 60 Gy was planned to be delivered to 98% of the intermediate-risk volume.

Patients were evaluated every week during CRT, every 2 weeks after CRT completion while undergoing nivolumab maintenance administration, and every 12 weeks thereafter. Blood samples were collected at baseline and during treatment. A tumor biopsy was performed at baseline only. Radiological assessment was performed at baseline, at brachytherapy initiation (week 7–8), at week 14–16, at week 25, and every 6 months thereafter by both pelvic MRI (T2- and T1-weighted images with gadolinium-chelates enhancement, and diffusion-weighted sequences) and 18F-fluorodeoxyglucose ($^{18}$F-FDG) positron emission tomography/computed tomography (PET-CT). The study design is provided in Fig. 1A.

## HPV typing

Total DNA isolated from formalin-fixed tissue blocks was used for HPV typing. Real-time PCR using Sybr®Green and specific primers for HPV16, 18, and 33, was performed on a 7900HT Fast Real-Time PCR System (Applied Biosystems, Foster City, CA). Multiplexed amplification was done in a 25 µl volume using SYBR Green PCR Master Mix at

the final concentration 1X, HPV16 primers at 0.7 µM each, HPV18 and 33 primers at 1 µM each, DNA template (up to 100 ng) and nuclease-free water. HPV negative status was confirmed by PCR using consensus GP5+/GP6+ primers, which can detect a large spectrum of HPV types[66]. Sanger sequencing using GP5+ primer was performed on the PCR product to identify HPV type in patients with HPV16/18/33 negative status.

## Tumor tissue dissociation

Freshly resected tumor samples ($n = 10$ of which $n = 6$ for PF and $n = 4$ for PD) from untreated locally-advanced cervical cancer patients were cut in small fragments and enzymatically digested at +37 °C in agitation, during 30–45 min, in $CO_2$-independent medium (Gibco) supplemented with 5% of human serum (HS), Collagenase I (2 mg/mL) (Sigma), Hyaluronidase (2 mg/mL) (Sigma), and DNAse (25ug/mL) (Roche). Single-cell suspensions were filtered over a 40-µm cell-cell strainer (Fisher Scientific), washed at first with $CO_2$-independent medium + 5% HS and then washed with PBS 1x (Gibco) supplemented with 2 mM EDTA (Gibco) and 1% HS. Cell pellets were obtained following 5 min centrifugation at +4 °C. Cells were resuspended in $CO_2$-independent medium + 5% HS, counted and immediately stained for multiparametric flow cytometry (FC) analyses.

## Tumor-infiltrating lymphocytes (TILs) assessment

The density of TILs was determined based on the recommendation by the International TILs Working Group[67] ($n = 15$ eligible for analyses, of which $n = 11$ for PF and $n = 4$ for PD). We selected the tumor areas were selected at low magnification and assessed the percentage of the area that was filled with mononuclear cells in the stromal area around the tumor border at high magnification (×200). We defined all the mononuclear cells, including the lymphocytes in the stromal area, as TILs and excluded granulocytes and other polymorphonuclear leukocytes.

## Multiplexed immunohistochemistry

Paraffin-embedded tissue blocks ($n = 15$ eligible for analyses, of which $n = 11$ for PF and $n = 4$ for PD) were cut with a microtome into fine slivers of 3 microns. Immunostaining was processed in a Bond RX automated (Leica) with Opal™ 7-Color IHC Kits (Akoya Biosciences, NEL821001KT) according to the manufacturer's instructions. The multiplex panels consisted of the following antibodies: panel 1: CD3 (polyclonal), Granzyme B (clone GrB-7), FOXP3 (clone 236 A/E7), CD11c (clone 2F1C10), PD-L1 (clone ZR3), Cytokeratin (clone AE1/AE3); panel 2: Granzyme B (clone GrB-7), FOXP3 (clone 236 A/E7), CD11c (clone 2F1C10), Ki67 (clone MIB-1), Cytokeratin (clone AE1/AE3); panel 3: CD8 (clone C8/144B), CD28 (clone EPR22076), CD86 (clone EP1158-37), PD-1 (clone EPR4877(2)), CD11c (clone 2F1C10), Cytokeratin (clone AE1/AE3). Details regarding antibody dilutions, catalog number, and validations are provided in Supplementary Table 2. Tissue sections were coverslipped with Prolong™ Diamond Antifade Mountant (Thermo Fisher) and stored at 4 °C. Subsequently, slides were scanned using the Vectra® 3 automated quantitative pathology imaging system (Vectra 3.0.5; Akoya Biosciences). Multispectral images were unmixed and analyzed using the inForm Advanced Image Analysis Software (inForm 2.6.0; Akoya Biosciences). Nearest neighbor distances are performed using the R-script package phenoptr Reports (v0.3.2, Akoya BioSciences).

## Peripheral blood mononuclear cell isolation

Peripheral blood (PB) samples were collected at different time points: baseline ($n = 10$, of which PF = 7, PD = 3; six blood samples have not been collected), week 3 ($n = 13$, of which PF = 9, PD = 4; three blood samples have not been collected) and week 6 ($n = 13$, of which PF = 9, PD = 4; three blood samples have not been collected) post-initiation treatment. PB mononuclear cells (PBMCs) were isolated by density gradient centrifugation using Lymphoprep solution (StemCell),

collected in RPMI-1640 (Gibco) supplemented with 5% HS and 1% Penicillin/Streptomycin (Gibco) and immediately stained for flow cytometry analyses.

## Flow cytometry antibodies

Both tumor cell suspensions and PBMCs were stained with two different panels of antibodies ("Lymphoid" and "Myeloid" panels).

For the tumor lymphoid panel, the following mouse-anti-human antibodies were used: anti-CD3 (AF700, clone: UCHT1, Biolegend), anti-CD4 (BV650, clone: OKT4, Biolegend), anti-CD8 (PE-Texas Red, clone: 3B5, TermoFisher Scientific), anti-CD25 (BV605, clone: BC96, Biolegend), anti-CD27 (PerCP-Cy5.5, clone: M-T271, Biolegend), anti-CD45 (APC-Cy7, clone: 2D1, BD), anti-CD45RA (BV786, clone: HI100, BD), anti-CD56 (BUV395, clone: NCAM16.2, BD), anti-CD127 (AF488, clone: A01905, Biolegend), anti-ICOS (BUV737, clone: DX29, BD), anti-OX40 (PE, clone: Ber-ACT35, Biolegend), and anti-PD1 (AF647, clone: EH12.2H7, Biolegend).

For the blood lymphoid panel, the following surface antibodies were used: anti-CD3 (AF700, clone: UCHT1, Biolegend), anti-CD4 (BV650, clone: OKT4, Biolegend), anti-CD8 (PE-CF594, clone: 3B5, Thermo Fisher), anti-CD25 (BV605, clone: BC96, Biolegend), anti-CD27 (PerCP-Cy5.5, clone: M-T271, Biolegend), anti-CD45RA (BV786, clone: HI100, BD), anti-CD56 (BUV395, clone: NCAM16.2, BD), anti-ICOS (BUV737, clone: DX29, BD), and anti-OX40 (PE, clone: Ber-ACT35, Biolegend). For intracellular detection of Ki67 (BV421, clone: Ki67, Biolegend), the FOXP3 fixation-permeabilization kit was used (eBioscience). For PD-1 detection at baseline, cells were directly stained with anti-PD1 (AF647, clone: EH12.2H7, Biolegend). At week 3 and week 6, cells were first incubated with an anti-IgG4 secondary antibody (Biotin, clone: HP-6025, Sigma), recognizing the Fc portion of Nivolumab, and then with AF647- conjugated Streptavidin (Biolegend).

The following antibodies were used as isotype controls: BUV737 Mouse IgG1, κ (clone: X40, BD), and PE Mouse IgG1, κ (clone: MOPC-21, Biolegend).

For the tumor myeloid panel, the following mouse-anti-human antibodies were used: anti-CD3 (AF700, clone UCHT1, Biolegend), anti-CD11c (BUV395, clone: B-ly6, BD), anti-CD14 (FITC, clone: M5E2, BD), anti-CD16 (BUV737, clone: 3G8, BD), anti-CD19 (AF700, clone: HIB19, Biolegend), anti-CD45 (APC-Cy7, clone: 2D1, BD), anti-CD56 (AF700, clone: B159, BD), anti-CD141 (PerCP-Cy5.5, clone: M80, Biolegend), anti- CD163 (AF647, clone RM3/1, Biolegend), anti-CD206 (PE-CF594, clone 19.2, BD), anti-EpCAM (BV786, clone 9C4, Biolegend), anti-HLA-DR (BV650, clone: L243, Biolegend), anti-ICOS-L (PE-Cy7, clone: 2D3, Biolegend), and anti-PD-L1 (BV421, clone MIH1, BD).

For the blood myeloid panel, the following mouse-anti-human antibodies were used: anti-CD1c (BV421, clone: L161, Biolegend), anti-CD3 (APC-y7, clone: UCHT1, Biolegend), anti-CD11c (BUV395, clone: B-ly6, BD), anti-CD14 (AF488, clone: M5E2, BD), anti-CD16 (BUV737, clone: 3G8, BD), anti- CD19 (APC-Cy7, clone HIB19, Biolegend), anti-CD56 (AF700, clone: B159, BD), anti-CD123 (AF647, clone: 6H6, Biolegend), anti-CD141 (BV786, clone: M80, Biolegend), anti-HLA-DR (BV650, clone: L243, Biolegend), anti-ICOS-L (PE-Cy7, clone: 2D3, Biolegend), and anti-PD-L1 (PerCP-eFluor710, clone MIH1, eBioscience).

The following antibodies were used as isotype controls: PE-Cy7 Mouse IgG2b, κ (clone: MOPC-11, Biolegend), PerCP-eFluor710 Mouse IgG1, κ (clone: P3.6.2.8.1, eBioscience), and BV421 Mouse IgG1, κ (clone: MOPC-21, Biolegend).

In all panels, PBS(1×) + 2% FBS-diluted LIVE/DEAD™ Fixable Aqua Dead Cell Stain Kit (Thermo Fisher Scientific) was used for dead cell exclusions, according to the manufacturer's instructions. For both tumor and peripheral blood myeloid panels, a 10 min incubation at +4 °C with 1:25 PBS(1x)-diluted Fc receptor binding inhibitor (eBioscience) was performed, to prevent aspecific antibody binding of myeloid cells, prior to the direct addition of the antibody mix. The samples were acquired on an LSR Fortessa Cytometer (BD) and

analyzed with FlowJo software (TreeStar, version 10.8.0). Details regarding antibody dilutions, catalog numbers, validations, and other FC reagents are provided in Supplementary Table 3.

## Targeted DNA sequencing and bioinformatics' processing

The DNA sequencing was performed using the in-house generated Next-Generation Sequencing (NGS) panel DRAGON (Detection of Relevant Alterations in Genes involved in Oncogenetics), comprising 571 genes. To generate indexed paired-end libraries of tumor DNA, the Agilent SureSelect XT2 library prep kit (Agilent Technologies, Santa Clara, CA) was employed, enabling targeted sequencing of regions of the genome spanning 2.7 Mb. The library construction followed the manufacturer's protocol, using around 100 ng of input DNA. The sequencing was performed on a NovaSeq 6000 (Illumina) S2x150 bp flow cell (Illumina Inc., San Diego, CA).

Read mapping: in the initial analysis, reads were aligned to the human reference genome (hg19 assembly) using 'BWA' mem software[68] (v0.7.15) with default parameters. Quality control metrics for mapping, including the percentage of aligned reads (total and falling into the capture), PCR duplicates, as well as capture coverage, were obtained using a combination of "SAMtools flagstat", "BEDtools coverage", and "PicardTools MarkDuplicates". Subsequently, variant calling (both single-nucleotide variants (SNVs) and small insertion/deletion (indels)) was conducted on the processed alignment files using SAMtools mpileup[69] and *VarScan2* *mpileup2cns* (v2.4.3)[70]. To annotate small variants, several databases provided by ANNOVAR[71] were used: RefSeq, dbsnp v150, COSMIC v86, 1000 g project 08/2015 version, ESP6500, gnomAD (all and ethnicities), ICGC v21, and dbnsfp v35 predictions. Intermediate indels were annotated using the RefSeq database only. Variants within ±10 bp of each exon junction were classified as splicing events. A stringent selection algorithm was applied to filter out irrelevant variants, considering a minimum allelic ratio of 5% and a maximum frequency of 0.1% in the population. Truncating mutations (frameshift deletion and insertion, stopgain, splicing alteration, and hotspot mutations from the Cancer Hotspot database) in tumor suppressor genes were retained if they had a minimum coverage of 200 reads, while all missense variants known as hotspot mutations from the Cancer Hotspot database were retained for oncogene variants, regardless of coverage. For genes classified as both oncogenes and tumor suppressor genes (such as NOTCH1) or with known missense hotspots (like TP53), truncating mutations (minimal coverage of 200) and known hotspot mutations (with no minimal coverage) were selected. The tumor mutational burden (TMB) was calculated as the number of non-synonymous somatic mutations (SNVs and small indels) per megabase in coding regions (mut/Mb). Only coding variants (except for intronic splicing ones, therefore exons-only, representing 1.59 Mb), without synonymous variants or polymorphisms (>0.1% minor allele frequency) and recurrent variants covered enough (not tagged as Low_Depth) were considered in the TMB calculation. Microsatellite Instability (MSI) was assessed using MSIsensor2 (https://github.com/niu-lab/msisensor2, commit ebdbf42, niu-lab) with an MSI score cut-off of 15% to consider MSI status. Manual curation was also performed to validate the MSI status of those samples. Oncoprints were drawn using the ComplexHeatmap package and were carried out with the Maftools package for the 4.00 version of R.

## RNA extraction, sequencing, and data processing

Total RNA was extracted from FFPE sections ($n = 11$ for PF, $n = 4$ for PD) using the high pure FFPET RNA isolation kit (Roche, Basel, Switzerland) according to the manufacturer's protocol. RNA yield and quality was determined with the NanoDrop™ One spectrophotometer (Thermo Fisher Scientific, Waltham, MA, USA) and fragment size was analyzed using the RNA ScreenTape assay run on the 4200 Bioanalyzer (Agilent Technologies, Santa Clara, CA, USA). DV200 values representing the percentage of RNA fragments above 200 nucleotides in length were

estimated, and cases with DV200 of more than 30% were included for library preparation.

Libraries were prepared from 120 ng of total RNA using the Illumina TruSeq RNA Exome Library preparation kit which allows to perform a strand-specific RNA sequencing. After individual QC, the 16 libraries were equimolarly pooled and subjected to qPCR quantification using the KAPA library quantification kit (Roche). Sequencing was carried out on the NovaSeq 6000 instrument from Illumina based on a 2 × 100 cycle mode (paired-end reads, 100 bases) to obtain around 25 million clusters (50 million raw paired-end reads) per sample. Fastq files were generated from raw sequencing data using bcl2fastq where demultiplexing was performed according to barcodes.

RNA-seq data were processed using the Institut Curie RNA-seq pipeline[72] (Zenodo, v3.1.8 https://github.com/bioinfo-pf-curie/RNA-seq). Briefly, after trimming of the adapter sequences, the reads were aligned on the Human reference genome (hg38) using the STAR software (v2.6.1). The gene expression matrix was generated with STAR[73] using the quanMode Gene Counts option. Coding genes (annotated with Gencode v29) with very low expression levels (<1 TPM on all samples) were filtered out from the analysis. Counts data were then normalized using the trimmed mean of M-values (TMM) method of the edgeR package[74]. Differential expression was assessed between PF and PD samples with the R package Limma-voom[75]. No gene was differentially expressed with an adjusted $p$ value threshold of 0.05. Gene set enrichment analysis (GSEA) and gene set variation analysis (GSVA) were performed using respectively the fgsea R package[76] and the Bioconductor GSVA package[77] on MSigDB (v7.5.1) Hallmark gene sets[78]. For GSEA analysis, gene were pre-ranked according to the assigned differential expression t-score. Gene sets related to epithelial-to-mesenchymal transition (EMT), angiogenesis, *KRAS* signaling, *interferon-(IFN-) alpha*, and *IFN-gamma* were among the top significantly enriched gene sets. Differential analysis of GSVA enrichment score between PF and PD samples have been performed with the Limma R package (v3.54.0). No GSVA gene set was statistically significant after $p$ value correction with Benjamini–Hochberg. Volcano plots and the GSEA were drawn using ggplot2 (v3.4.0) and ggrepel (v0.9.2) package. The GSVA plot was drawn using the ComplexHeatmap package (v2.14.0).

**Reporting summary**

Further information on research design is available in the Nature Portfolio Reporting Summary linked to this article.

## Data availability

The raw and processed data from both targeted DNA and RNA sequencing have been deposited in the EGA database under accession code EGAS00001007297. The data were available under restricted access after review by the Data Access Committee (DAC) because these are human data generated from medical research. Request for data access will be referred directly to our DAC at data.request@curie.fr. Individual, de-identified, participant sequencing, and clinical data will be accessible for research purposes only as specified in the data access policy. After review of the request, the data access decision will be passed to the EGA database within four weeks, and an access account will then be granted. In case of publication(s) from the requester, the data will be available for 2 years from the date of last publication. If there is no use of the data for a period of 2 years, the requester must delete the data. The study protocol is available as Supplementary Note in the Supplementary Information File. The remaining data are available within the Article, Supplementary Information, or Source Data file. Source data are provided with this paper.

## Code availability

RNA-seq data were processed using the Institut Curie RNA-seq pipeline[72] (v3.1.8, https://doi.org/10.5281/zenodo.7446922).

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

## Acknowledgements

We thank all the platforms and services at Curie involved in the study: Cytometry, Experimental Pathology, and Clinical Immunology. Celine Cousin for technical contribution on FACS staining. High-throughput sequencing was performed by the ICGex NGS platform of the Institut Curie supported by the grants ANR-10-EQPX-03 (Equipex) and ANR-10 INBS-09-08 (France Génomique Consortium) from the Agence Nationale de la Recherche ("Investissements d'Avenir" program), by the ITMO-Cancer Aviesan (Plan Cancer III) and by the SiRIC-Curie program (SiRIC Grant INCa-DGOS- 4654). We thank Alexia Savignoni for input on the trial design. We thank Pierre-Emmanuel Bonté for his advice on biostatistical analyses. We thank Nina Jehanno and Christophe Le Tourneau for their support with patient care. We thank Charlotte Martinat and Constance Lamy for their help accessing archived material. We thank Sabrina Leclercq, Marie-Emmanuelle Legrier, Jennifer Dieppedale, and Anne-Sophie Plissonnier for support with data acquisition, monitoring, and management of trial procedures. This work was supported by the following grants to ER: Bristol-Myers Squibb (BMS) funded the clinical trial and the associated translation research. BMS had no role in study design, data collection and analysis, or manuscript writing. CIC IGR-Curie 1428; ANR-10-IDEX-0001-02 PSL; ANR-11-LABX-53 0043. M.R. is supported by the Interface INSERM program. E.T. was supported by a postdoctoral fellowship abroad from the AIRC (2018/2020-54 number: 20934).

## Author contributions

M.R. and E.R. conceived and designed the trial, coordinated trial procedures, analysed, and interpreted the clinical data, and wrote the manuscript. M.R., E.R., Co.D. included and treated patients in the trial. P.L. analysed and interpreted clinical data and wrote the manuscript. E.R. conceived and supervised the immune monitoring experiments. G.V. analysed and interpreted translational data and wrote the manuscript together with E.T. E.T. contributed to FACS staining and analysis. M.M., L.B., and Ca.D. included patients in the trial and delivered radiation therapy. V.F., E.L., and N.P. included patients and performed the majority of biopsies. La.L. performed the histological analyses. D.M. and A.V.-S. supervised the histological analyses. S.V. performed NGS and transcriptomic sequencing. Lo.L. performed the bioinformatic analyses of NGS and transcriptomic sequencing. N.S. and I.B. supervised the NGS and transcriptomic sequencing and the related bioinformatic analyses. M.C. performed the statistical analysis of the clinical data. C.M., V.H., and L.C. performed the assessment of pelvic MRI and PET-CT scans. Z.C.-A. and G.M. helped with trial coordination. M.K. gave input on trial design, coordinated tissue biobanking, and gave input on NGS and transcriptomic sequencing analyses. S.A. and O.L. advised on the immune monitoring analyses. All authors edited and approved the manuscript.

## Competing interests

M.R. reports personal fees for serving as an advisor from Merck Sharp & Dohme, AstraZeneca, GlaxoSmithKline, Immunocore; travel support from AstraZeneca; funds to his institution to support a study from Merck Sharp & Dohme. E.R. reports investigator-initiated trial funds (paid to the institution) by AstraZeneca, BMS, and Replimune; serves on the consultancy/advisory board for AstraZeneca, Merck, Roche, Light Chain Biosciences, Pierre Fabre; E.R. declares travel support from BMS, Hoffmann La Roche, AstraZeneca, Merck Sharp & Dohme. G.V. and E.R. received grants from the Fonds Amgen France pour la Science et l'Humain. MR and ER report funding to their institution to support a study from Janssen-Cilag. CaD received grants from AstraZeneca, Janssen, and Astellas. M.K. received funds from Roche. The remaining co-authors declare no competing interests.

## Additional information

Manuel Rodrigues [1,14], Giulia Vanoni[2,14], Pierre Loap[3], Coraline Dubot[1], Eleonora Timperi [2], Mathieu Minsat[3], Louis Bazire[3], Catherine Durdux[4], Virginie Fourchotte[5], Enora Laas[5], Nicolas Pouget[5], Zahra Castel-Ajgal[6], Gregoire Marret[6], Laetitia Lesage [7,8], Didier Meseure [7,8], Anne Vincent-Salomon [7,8], Lolita Lecompte [9], Nicolas Servant [9], Sophie Vacher [10], Ivan Bieche[10], Caroline Malhaire [11], Virginie Huchet[12], Laurence Champion [12], Maud Kamal[6], Sebastian Amigorena [2], Olivier Lantz [2], Marion Chevrier[13] & Emanuela Romano [1,2] ✉

[1]Department of Medical Oncology, Institut Curie, Paris & Saint-Cloud, France. [2]Center for Cancer Immunotherapy, INSERM U932, PSL Research University, Institut Curie, Paris, France. [3]Department of Radiation Oncology, Institut Curie, Paris & Saint Cloud, France. [4]Hôpital Européen Georges Pompidou, Department of Radiation Oncology, Paris, France. [5]Service of Breast and Gynecologic Surgery, Institut Curie, Paris, France. [6]Department of Drug Development and Innovation, Institut Curie, Paris, France. [7]Department of Pathology Institut Curie, Paris, France. [8]Centre d'Investigation Clinique Biothérapie, Institut Curie, Paris, France. [9]Institut Curie Bioinformatics Platform, INSERM U900, Mines ParisTech, Paris 75005, France. [10]Pharmacogenomics Unit, Service of Genetics, Institut Curie, Paris, France. [11]Department of Radiology, Institut Curie, Paris 75005, France. [12]Department of Nuclear Medicine, Institut Curie, Paris 75005, France. [13]Service of Biostatistics, Institut Curie, Paris 75005, France. [14]These authors contributed equally: Manuel Rodrigues, Giulia Vanoni. ✉e-mail: Emanuela.Romano@curie.fr

