## [Peer Review File · Nature Communications]

Nivolumab plus chemoradiotherapy in locally-advanced cervical cancer: the NICOL phase 1 trialREVIEWER COMMENTS

Reviewer #1 (Remarks to the Author): with expertise in cervical cancer, immunotherapy

Thank you for the chance to review this interesting work. In this single arm phase 1 study, 16 patients with locally advanced cervical cancer were treated with cis-RT with concurrent nivolumab, followed by nivolumab maintenance. Multiparameter immune monitoring was performed, including tumor microenvironment analysis and peripheral flow cytometry.

This paper contributes to the field in regards to the cervical tumor microenvironment as well as peripheral changes in the immune compartment in response to cis-RT plus nivolumab. My comments are as below:

1. In the methods discussion, would be helpful to note how often the scans were performed for the endpoint of ORR.
2. The way that the efficacy is described is a bit confusing. My interpretation is that 15 out of the 16 patients had tumor shrinkage at the time of landmarked scan 8 weeks after brachytherapy. One patient had POD at that time and went to surgery. An additional 3 patients had progression at a later date. Those 4 patients are the “non responders” in your analysis, but that’s not exactly accurate as they achieved a response by RECIST, and then had progression. It may be more accurate and clearer to describe the two groups as those who are progression-free, and those who had POD.
3. Because of the amount of translational data, it would be helpful to have a supplemental figure to understand what the overlap in the datasets are. For example, a venn diagram of how many patients had NGS, PBMC, and mIF data.
4. It’s notable that 2 patients in the cohort appeared to have HPV negative tumors. Were they both endocervical adenocarcinoma? Were they both patients that had progression? These might be gastric-type tumors, which are less radiosensitive and have a more aggressive phenotype and may have influenced your outcomes. Please address in the discussion.
5. In the discussion, it’s an important to acknowledge that the CALLA study was negative, and this study is much smaller. The ORR is likely comparable and within the HR of the CALLA study given the small numbers.
6. It’s also important to acknowledge in the discussion that this trial has no comparator arm – the presence of enhanced TIL in the pre-treatment biopsies of responders may be more prognostic of disease behavior and less predictive of response to nivolumab. It’s impossible to know without a placebo control.
7. Figure 2A is not projecting well – I don’t see any grey or orange dots as described in the legend.
8. The p values seem off in figure 3 panels C,D,G – aren’t you comparing the responders and non responders? The p values seem to be comparing the responders in the tumor vs the stroma
9. If this is the primary study publication, it would be good to include a table with patient demographics, and more information about AEs beyond high grade AEs/DLTs. These could be supplemental tables.
10. The figures are a little busy, and it can be hard to derive what is significant from the way they are designed. The authors might benefit from moving more of the non-significant data to supplemental, and focusing on the findings that are significant and that they highlight in the paper.

Reviewer #2 (Remarks to the Author): with expertise in cervical cancer, immunotherapy, radiotherapy

The authors present the phase I and TR results of the NiCOL trial, which included 16 women with LACC (Ib3-IVa) who were treated with chemo-radiation and 240 mg flat dose nivolumab.

The study is well-designed and follows current best practices (external beam radiation + brachy + cisplatin weekly 40mg/m²).

The nivolumab flat dose + cisplatin + RT combination required phase I testing at the time the study was designed, so the question was relevant, and the phase I trial was properly designed.

Treatments were given correctly, and the reported compliance indicates that the trial and patient care were of high quality.

Furthermore, recent phase III trials found that immunotherapy, when combined with chemo and bevacizumab, was beneficial in the metastatic setting but not in the primary setting in patients with locally advanced disease when combined with cisplatin-radiation.

The authors report an overall response rate of 93.8% and a 2-year PFS of 75%, which is consistent with previous results with cisplatin-radiation and proper brachytherapy delivery.

Overall, this study's extensive efforts to depict immune activation profiles prior to CRT-immunotherapy treatment may well have clinical translational value.

However, significant issues must be addressed.

For example, there is a lack of granularity in immune infiltration and gene expression in various group settings (for example, those with a poor prognosis per se, such as pelvic and/or para-aortic positive patients).

Similarly, it is difficult to reach a consensus on which PD-L1 cut-off level should be used for future clinical trial design as the numbers are too small and the authors do not provide this information.

The study could be improved in order to provide a clear understanding of the immune biology behind cervical cancer.

Introduction

1. Line 82: "However, optimally-treated LACC is still associated with a 40% risk of disease recurrence (Ref 6)"

It is true that some studies show 40% recurrence, but the GEC-ESTRO group is probably the one with the best results in Europe and should be mentioned.

Results

1. In the patient characteristics it would be important to add the number and percentage of patients with positive, negative or equivocal pelvic lymph nodes and para-aortic lymph nodes, as well as tumor grade (grade1-2 vs grade 3) as these are well described adverse prognostic factors in LACC.

2. Line 366: adjuvant surgery was conducted in three patients (18.75%), at week 15 for two patients and at week 23 for one patient. As the authors had the hysterectomy specimen it would be very important that they perform correlative studies in the tumours or lymph nodes of the surgical specimen. Were the recurrent tumors infiltrated by "M2" macrophages? Tregs? Genomic signatures?

3. The Figure 1 could be enriched with a swimmer's plot indicating which were the patients that progressed, timepoints of progression, toxicity and interruptions of treatment, etc. Similarly Figure 1 could be enriched with clinical vignettes of responder patients.

4. Figure 1A should indicate the timepoint when the biopsies were taken. Please clarify if the patients had paired biopsies. If this was not possible then please recognise it in the text.

5. I would suggest enriching the paper with a Supplementary Table with the description and distribution of all adverse events by grade and MedDRA classification. Also please expand on the description of the phase I study. Was it a safety run in? How decisions were taken? Was there an IDSMC implicated? Were there any stopping rules if immune related toxicity appeared?
6. Line 439 At the DNA level, using a dedicated NGS gene panel, the main observed alterations across all patients were in FAT1 (30%), STK11 (30%), CASP8 (20%), PIK3CA 441 (20%) and YAP1 (20%). No KRAS, NRAS or HRAS mutation was found. Could the authors please give references and description of the function of those genes? Do the authors can generate any hypothesis on what is the significance of the presence of those genes? Were these genes previously described in the literature of cervical cancer?
7. The same institution published recently the results of a translational study where by NGS identified PIK3CA mutations as the most frequent oncogenic alterations. What are the similarities and differences in the signatures that the authors are showing in this paper vs the previous paper published by the same institution.
<https://www.sciencedirect.com/science/article/pii/S2352396419302142?via=ihub>
As this previous paper included 182 patients, could that paper be used as a validation of the signatures? I think this could be of great value for the medical community.
8. Are the genes found to be the most cervical cancer-specific in the pan-cancer Cancer Genome Atlas (TCGA)? Signatures like PTEN (Peng et al Cancer Discovery 2015), PI3K (Ali et al Nature 2014;510:407–11), B-catenin (Spranger et al Nature 2015) are important correlates of response and it would be interesting to understand why these signatures are not there? Was it because of the small number of patients? Have the authors looked into that?
9. Similarly, recent papers suggest that cervical tumors harbor HR deficiency
<https://www.ncbi.nlm.nih.gov/pmc/articles/PMC8606581/> which usually harbors higher number of T cells. CNV (amplification and deletions together) could indicate genomic instability which are usually linked to higher immunogenicity due to cGAS/STING signaling. Would have been nice to incorporate pSTAT1 and pSTAT3 as correlative markers of type II and type I IFN.
10. Figure 2: The non-responder patients have signatures of angiogenesis and EMT as well as reduced IFN type I and type II response. These signatures should be properly described with volcano plots of the most differentially expressed genes for responders and non-responders.
11. Figure 3 and text under “Brisker pre-existing tumor infiltrating immune infiltrate at baseline in responder patients”. Please in the text avoid using terms such as “slight”, “trend”, “difference”, “higher”...replace it by statistically significant or non-statistically significant. Also, and this is valid for all the Figures when the p-value is not statistically significant either add the p-value in numbers or add NS p-value (every column comparison should have a p-value)
12. Figure 3 mIF images should be of better quality. Epithelial to mesenchymal transition was one of the signatures observed by RNAseq. The authors mentioned that there are differences between stroma and tumor immune infiltration, but I cannot see a specific marker for stroma, was it done?
13. Line 455 Under: “Non-tumoral PD-L1+ cells - comprising immune and stromal cells - significantly accumulated in the stroma of R patients, along with a significant increase of PD-L1+ tumor cells (expressing cytokeratins, CKs) in the tumor areas of R patients (Fig. 3D and Extended Data Fig. 2B)
14. This study's findings will be useful in the design of future clinical trials. For example, this study could help future investigators determine the PD-L1 cut-off as inclusion criteria in future trials. It is thus critical to understand which assay was used to determine PD-L1 staining. Typically, the approved assay for nivolumab clinical trials is antibody 28-8, with a threshold of >1% on TC. Because of the small number of patients and response rates to CRT+N that are very similar to those published in the literature with CRT alone, drawing conclusions will be difficult, but because this is the only

information we could eventually have it would be of value if the authors include this in the paper.

15. There is statistically significant difference in the proportion of CD3 cell in the stroma in responders vs non-responders. Can the authors look at the presence of CD8+PD1+ cells ? These are probably the cells that express GrzB and Ki67+ which can be the effectors (exhausted T cells). It would be important to look at the ratio of CD8+PD1+/CD4+Foxp3+ cells.

16. Overall, their findings support the idea that T-cell activation and cytotoxicity can be used to predict anti-PD1 success. Their findings also suggest that preexisting immunogenic and inflamed malignant cells with the ability to generate tumor reactive TILs in situ are required for success. As recent papers show that CD11b+ or CD11c+ cells occupy niches in close proximity to T cells expressing PD1, it is important to understand whether this proximity implies more antigen presentation and co-stimulation capacity from the CD11b+ or CD11c+ cells in cervical tumors, which could also explain more exhausted (PD1+ T cells). Can the authors supply CD80, CD86, and CD28? Due to the lack of CD28 costimulatory cues, it is possible that solitary PD-1+CD8+ TIL are more likely to reach dysfunctional exhausted states in situ. If this is the case, the authors could go back to the DNA sequencing data and look at the signatures that patients with properly costimulated T cells have.

17. Recent papers show that the irradiation of the tumor draining lymph nodes abrogates immunological responses. Please see <https://www.nature.com/articles/s41591-022-02084-8>. Supp Fig 7h in this paper: Patient who received 25x1.8 Gy on para-aortal lymph nodes alongside the spine 12 months before the 89ZED88082A PET scan. See also <https://www.nature.com/articles/s41467-022-34676-w> See also <https://pubmed.ncbi.nlm.nih.gov/29898992/>

These papers should be commented as they derive important lessons for the future where we should probably be more economic in the irradiation of lymph node areas where priming takes place.

Particularly knowing that CD8+ T cell activation in cancer comprises an initial activation phase in lymph nodes followed by effector differentiation within the tumor

<https://doi.org/10.1016/j.immuni.2022.12.002>

Would the authors propose any solution or hypothesis generating idea in order to tackle this problem?

Could the authors re-analyse the data by separating patients based on para-aortic positive disease or para-aortic irradiation? Did these patients had more immune suppressive signatures at baseline? Do they had more Tregs? More M2 macrophages? How did they respond to treatment? The numbers maybe small, and thus the importance of validating this cohort with other cohorts of patients to derive meaningful conclusions.

Discussion

1. Line 559 "We observed a 25% rate of acute, grade 4 lymphopenia, likely related to a synergistic effect between cisplatin, radiotherapy and nivolumab". I could not find this information in the paper and I found it in the discussion.

Chronic lymphopenia has been described in patients with cervical cancer, correlations analysis using FDG-PET/CT showed that radiation reduces the active bone marrow justifying the chronic myeloid toxicity observed. I would suggest that the authors include their toxicity in the results section and that the lymphopenia is depicted at baseline and at subsequent measurements. In oncology during follow up we perform blood work across several months for these patients, the authors well recognised this in line 567 "These observations should justify repeated laboratory tests". If the authors have the data, it would be important to see baseline and subsequent measurements. For example, it would be important to depict WBC, neutropenia, lymphopenia. This could also enrich Figure 1.

2. The discussion should recognize the short follow-up of the patients which explain why the PFS and LC are so good. Maybe if the data is re-analysed with longer follow-up more non-responding patients will appear.

3. Although the discussion is clinical, the entire paper is translational. The authors should better

discuss their findings and compare them to other papers on gynecological cancers (ovarian, endometrial cancer for instance). Are the TME elements described by the authors universal observations across other tumor types?

4. The discussion should recognize as a limitation that biopsies were only obtained at baseline. By analyzing the malignancies before, during and after treatment, investigators could have had a better understanding of the the interaction among the cancer, the treatment and the immune system.

5. Based on the reviewer's new experiments, the investigators could develop proposals for combinations of therapies that will almost certainly be the key to better patient outcomes. Which drugs to use in the future? The problem with immunotherapy today is that its use in patients has outpaced a fundamental understanding of how it works, and we will not reach that understanding unless researchers conduct intensive monitoring of tumors via repeated thoroughly interrogated biopsies.

Reviewer #3 (Remarks to the Author): with expertise in biostatistics, bioinformatics

Comments on the clinical trial study design and statistical analyses.

Rodrigues et al. performed a phase 1 clinical trial of nivolumab plus chemoradiotherapy for locally-advanced cervical cancer. The primary endpoint is the DLT occurrence rate within 11 weeks after the initiation of treatment. The study used a 3+3 design to confirm the dose of nivolumab - 240 mg flat dose q2wk. The dose of nivolumab was initially evaluated in 6 patients and then in additional 9 patients in the expansion cohort. The treatment was considered acceptable if less than 2 patients experiencing DLT in the first 6 DLT evaluable patients.

The 3+3 study design was common in phase 1 studies. The clinical trial data for the primary and secondary endpoints were summarized by descriptive statistics, which are common in reporting phase 1 clinical trial. The DLT rate was 20%, which was considered acceptable.

The patient samples were analyzed by DNA and RNA sequencing. The bioinformatic and statistical analyses are generally appropriate. The major weakness of this report is the small sample size. Among the 16 patients, 12 are responders and 4 are non-responders. The conclusions based on comparison of the 12 responders and 4 non-responders are not convincing due to the small sample size, especially for the non-responders. Some significant findings are found in the responders, but not in the non-responders that might be simply due to the small sample size (e.g., Figure 3 C, D, E, G). In addition, multiple hypotheses were tested for immune markers, but the p-values were not adjusted for multiple comparison. Many marginally significant comparisons would not be statically significant any more if the raw p-values are adjusted for multiple comparison. The results shown in Figures 3, 4 and 5 should be interpreted carefully.

Manuscript NCOMMS-23-00781-T - Point-by-point Reply

REVIEWER COMMENTS

Reviewer #1 (Remarks to the Author): with expertise in cervical cancer, immunotherapy

Thank you for the chance to review this interesting work. In this single arm phase 1 study, 16 patients with locally advanced cervical cancer were treated with cis-RT with concurrent nivolumab, followed by nivolumab maintenance. Multiparameter immune monitoring was performed, including tumor microenvironment analysis and peripheral flow cytometry.

This paper contributes to the field in regards to the cervical tumor microenvironment as well as peripheral changes in the immune compartment in response to cis-RT plus nivolumab. My comments are as below:

1. In the methods discussion, would be helpful to note how often the scans were performed for the endpoint of ORR.

R: We thank the Reviewer for the opportunity to clarify the imaging schedule, which has been reported in the methods, **lines 178-181** as follows:

Radiological assessment was performed at baseline, at brachytherapy initiation (week 7-8), at week 14-16, at week 25, and every six months thereafter by both pelvic MRI (T2- and T1-weighted images with gadolinium-chelates enhancement, and diffusion-weighted sequences) and 18F-fluorodeoxyglucose (¹⁸F-FDG) positron emission tomography/computed tomography (PET-CT).

2. The way that the efficacy is described is a bit confusing. My interpretation is that 15 out of the 16 patients had tumor shrinkage at the time of landmarked scan 8 weeks after brachytherapy. One patient had POD at that time and went to surgery. An additional 3 patients had progression at a later date. Those 4 patients are the “non responders” in your analysis, but that’s not exactly accurate as they achieved a response by RECIST, and then had progression. It may be more accurate and clearer to describe the two groups as those who are progression-free, and those who had POD.

R: We thank the Reviewer for giving us the opportunity to clarify the efficacy data. We included a swimmer’s plot (**Figure 1B**) to better visualize the patient groups and, according to the Reviewer’s suggestion, we distinguished the subjects with progression-free as **PF**, and with progression of disease as **PD**, throughout the manuscript.

3. Because of the amount of translational data, it would be helpful to have a supplemental figure to understand what the overlap in the datasets are. For example, a venn diagram of how many patients had NGS, PBMC, and mIF data.

R: We thank the Reviewer for this useful suggestion. We have now included a Venn diagram (please, see below) in **Extended Figure 1A**, with the N of patients for whom different data types have been obtained and analyzed.

Extended Data Figure 1A. Nicol patient cohort and OncoPrint. (A) Venn diagram showing the overlap of patients across the different collected datasets. A total of 16 patients were included in the clinical trial. TILs assessment, mIF and Bulk RNA-seq were performed for 15/16 patients at baseline, comprising 11 PF and 4 PD patients. NGS was performed for all patients, but only 10/16 were of good quality, comprising 7 PF (6CR+1PR) and 3 PD. FC analyses were performed for 10/16 baseline tumors (comprising 6 PF and 4 PD patients). FC analyses of PBMCs were performed at baseline for 10/16 patients (7 PF, 3 PD), at week 3 and week 6 for 13/16 patients (9 PF, 4 PD).

4. It's notable that 2 patients in the cohort appeared to have HPV negative tumors. Were they both endocervical adenocarcinoma? Were they both patients that had progression? These might be gastric-type tumors, which are less radiosensitive and have a more aggressive phenotype and may have influenced your outcomes. Please address in the discussion.

R: The 2 patients in the cohort with HPV negative tumors of squamous cell carcinoma histology achieved complete response to treatment. Both patients had high baseline TILs (over 30%). We have addressed this point in the Result section "**Patients and treatments**".

5. In the discussion, it's an important to acknowledge that the CALLA study was negative, and this study is much smaller. The ORR is likely comparable and within the HR of the CALLA study given the small numbers.

R: Many thanks for raising this point. We have reformulated the discussion accordingly (**lines 614-619**).

6. It's also important to acknowledge in the discussion that this trial has no comparator arm – the presence of enhanced TIL in the pre-treatment biopsies of responders may be more prognostic of disease behavior and less predictive of response to nivolumab. It's impossible to know without a placebo control.

R: We thank the Reviewer for raising this point. In our study, TILs are similar in PF vs PD patients; however, we have reformulated the discussion (**lines 676-679**) pointing to caution in the interpretation of the biomarker data, in light of the absence of a comparator arm.

7. Figure 2A is not projecting well – I don't see any grey or orange dots as described in the legend.

R: Thank you for pointing this out. We will provide another version of the panel A in Figure 2. If the visualization problem persists, we will send a separate file at higher resolution.

8. The p values seem off in figure 3 panels C,D,G – aren't you comparing the responders and non responders? The p values seem to be comparing the responders in the tumor vs the stroma.

R: We thank the reviewer for raising this point. To not overload the figures, we opted to show only statistically significant comparisons in all figures. Indeed, in Figure 3, we only show statistical comparisons in the tumor vs the stroma of PF, since PF vs PD comparisons were not statistically significant.

9. If this is the primary study publication, it would be good to include a table with patient demographics, and more information about AEs beyond high grade AEs/DLTs. These could be supplemental tables.

R: We thank the reviewer for this suggestion. Indeed, it is useful to have more information about AEs. A table reporting treatment-related adverse events has been added as supplemental material (**Extended Table 1**). Patients' demographics are shown in Table 1.

10. The figures are a little busy, and it can be hard to derive what is significant from the way they are designed. The authors might benefit from moving more of the non-significant data to supplemental, and focusing on the findings that are significant and that they highlight in the paper.

R: We understand the Reviewer's comment; however, we felt that keeping some data – although non-significant – in the main figures could offer a broader overview on the immune landscape.

Reviewer #2 (Remarks to the Author): with expertise in cervical cancer, immunotherapy, radiotherapy

The authors present the phase I and TR results of the NiCOL trial, which included 16 women with LACC (Ib3-IVa) who were treated with chemo-radiation and 240 mg flat dose nivolumab. The study is well-designed and follows current best practices (external beam radiation + brachy + cisplatin weekly 40mg/m²). The nivolumab flat dose + cisplatin + RT combination required phase I testing at the time the study was designed, so the question was relevant, and the phase I trial was properly designed. Treatments were given correctly, and the reported compliance indicates that the trial and patient care were of high quality. Furthermore, recent phase III trials found that immunotherapy, when combined with chemo and bevacizumab, was beneficial in the metastatic setting but not in the primary setting in patients with locally advanced disease when combined with cisplatin-radiation.

The authors report an overall response rate of 93.8% and a 2-year PFS of 75%, which is consistent with previous results with cisplatin-radiation and proper brachytherapy delivery. Overall, this study's extensive efforts to depict immune activation profiles prior to CRT-immunotherapy treatment may well have clinical translational value.

However, significant issues must be addressed. For example, there is a lack of granularity in immune infiltration and gene expression in various group settings (for example, those with a poor prognosis per se, such as pelvic and/or para-aortic positive patients). Similarly, it is difficult to reach a consensus on which PD-L1 cut-off level should be used for future clinical trial design as the numbers are too small and the authors do not provide this information. The study could be improved in order to provide a clear understanding of the immune biology behind cervical cancer.

Introduction

1. Line 82: "However, optimally-treated LACC is still associated with a 40% risk of disease recurrence (Ref 6)"

It is true that some studies show 40% recurrence, but the GEC-ESTRO group is probably the one with the best results in Europe and should be mentioned.

R: We thank the Reviewer for this insightful comment. We have now mentioned in the Introduction section (**lines 83-85**) that the disease recurrence can be significantly improved by image guided brachytherapy, as demonstrated in RetroEMBRACE study.

Results

1. In the patient characteristics it would be important to add the number and percentage of patients with positive, negative or equivocal pelvic lymph nodes and para-aortic lymph nodes, as well as tumor grade (grade 1-2 vs grade 3) as these are well described adverse prognostic factors in LACC.

R: We thank the Reviewer for this proposal. In Table 1, we have reported the FIGO staging and the differentiation as follows: poorly, moderately, and well differentiated – following the advice of the pathologist, who reviewed the cases.

2. Line 366: adjuvant surgery was conducted in three patients (18.75%), at week 15 for two patients and at week 23 for one patient. As the authors had the hysterectomy specimen it would be very important that they perform correlative studies in the tumours or lymph nodes of the surgical specimen. Were the recurrent tumors infiltrated by “M2” macrophages? Tregs? Genomic signatures?

R: We fully agree with the Reviewer that the hysterectomy specimen would be of translational interest; however, the patients did not sign an informed consent to allow these types of analyses.

3. The Figure 1 could be enriched with a swimmer’s plot indicating which were the patients that progressed, timepoints of progression, toxicity and interruptions of treatment, etc. Similarly Figure 1 could be enriched with clinical vignettes of responder patients.

R: We thank the Reviewer for this useful suggestion. A swimmer’s plot has been included in **Figure 1B**.

4. Figure 1A should indicate the timepoint when the biopsies were taken. Please clarify if the patients had paired biopsies. If this was not possible then please recognise it in the text.

R: We apologise if the timepoint of tissue collection was unclear. The tissue biopsy was taken at baseline before treatment only. We have clarified in the text.

5. I would suggest enriching the paper with a Supplementary Table with the description and distribution of all adverse events by grade and MedDRA classification. Also please expand on the description of the phase I study. Was it a safety run in? How decisions were taken? Was there an IDSMC implicated? Were there any stopping rules if immune related toxicity appeared?

R: We thank the Reviewer for this suggesting that will increase the completeness of the data reported. A table all adverse events by CTCAE version 4.03. and MedDRA classification has been added as supplemental material (Extended Table 1). In the methods’ **“Study design, endpoints and statistical analysis”** section, we expanded on the description of the phase I study, which used a 3+3 design to confirm the dose of nivolumab - 240 mg flat dose q2wk (**lines 140-141**). The dose of nivolumab was initially evaluated in 6 patients and then in additional 9 patients in the expansion cohort. The treatment was considered acceptable if less than 2 patients experiencing DLT in the first 6 DLT evaluable patients. A Data Safety Monitoring Board was consulted to review the safety profile of the treatment in the of the first 6 DLT evaluable patients and in order to proceed with the expansion cohort.

6. Line 439 At the DNA level, using a dedicated NGS gene panel, the main observed alterations across all patients were in FAT1 (30%), STK11 (30%), CASP8 (20%), PIK3CA 441 (20%) and YAP1 (20%). No KRAS, NRAS or HRAS mutation was found. Could the authors please give references and description of the function of those genes? Do the authors can generate any hypothesis on what is the significance of the presence of those genes? Were these genes previously described in the literature of cervical cancer?

R: We thank the Reviewer for this comment. FAT1, STK11, CASP8, PIK3CA and YAP1 alterations were previously described in CC¹⁻³. This information and the functions of these genes in cancer is added to the discussion section, **lines 599-605**:

“FAT1, STK11, CASP8, PIK3CA and YAP1 were previously described in cervical cancer⁵³⁻⁵⁵ and were among the most recurrent alterations found in this study cohort. These genes encode for proteins known to be involved in cell proliferation and metastasis inhibition, regulation of cell polarity and metabolism, regulation of apoptosis, oncogenic transformation, and therapy resistance⁵⁶⁻⁶⁰. However, possibly due to the limited number of patients, we could not find any correlation between the presence of one of these alterations and ORR.”

FAT1, a member of the FAT cadherin superfamily, is a ~500-kD transmembrane protein that regulates actin dynamics, cell-cell adhesion, and cell polarity. FAT1 inhibits the proliferation and metastasis of cervical cancer cells by binding β -catenin⁴.

The most important PI3K pathway proteins are those that belong to class IA composed of a catalytic subunit p110 α (encoded by the *PIK3CA*), and its associated regulatory subunit p85 (encoded by the *PIK3R1*). PI3KCA catalyzes the formation of PIP3, a process that is reversed by the action of PTEN. PIK3CA mutations stimulate the oncogenic transformation via constitutive activation of p110 α , constitutive phosphorylation of AKT T308 and S473 and p70 S6 kinase⁵.

Serine/threonine kinase 11 (STK11) has been identified as a tumor suppressor gene, which inhibit mammalian target of rapamycin (mTOR) from the PI3K/AKT/mTOR Pathway. STK11 encodes for a serine/threonine kinase that regulates cell polarity and energy metabolism and functions as a tumor suppressor (Gene card). Mutations in LATS1, FAT1, JUB, STK11 or NF2 (result in aberrant activation of the downstream transcription factor, Yes1 associated transcriptional regulator (YAP1)³.

Casp8 is the upstream protease of the activation cascade of caspases responsible for the Tnfrsf6/FAS and Tnfrsf1A mediated cell death through the death-inducing signaling complex (DISC). Caspase-8 contributes to the functionality of the DNA damage response and has been originally identified as an essential player of apoptosis¹. CASP-8 LoF was associated with good prognosis in CC treated by chemoradiation².

YAP1 is a transcriptional co-activator whose activity is controlled by the Hippo signaling pathway. In addition to important functions in normal tissue homeostasis and regeneration, YAP1 has also prominent functions in cancer initiation, aggressiveness, metastasis, and therapy resistance⁶.

7. The same institution published recently the results of a translational study where by NGS identified PIK3CA mutations as the most frequent oncogenic alterations. What are the similarities and differences in the signatures that the authors are showing in this paper vs the previous paper published by the same

institution. <https://www.sciencedirect.com/science/article/pii/S2352396419302142?via=ihub>

As this previous paper included 182 patients, could that paper be used as a validation of the signatures? I think this could be of great value for the medical community.

R: We thank the Reviewer for making this point. Scholl *et al*² reported in the RAIDs study that loss of function in epigenetic acting genes together with driver alterations in the PI3K pathway were significantly associated with poor outcome in cervical squamous cell cancer. The previous study included 182 patients, who received standard treatment (radio-chemo therapy) and did not receive immunotherapy. As a consequence, the RAIDs population cannot be used a validation population for the signatures identified in the present study.

8. Are the genes found to be the most cervical cancer-specific in the pan-cancer Cancer Genome Atlas (TCGA)? Signatures like PTEN (Peng et al Cancer Discovery 2015), PI3K (Ali et al Nature 2014;510:407–11), B-catenin (Spranger et al Nature 2015) are important correlates of response

and it would be interesting to understand why these signatures are not there? Was it because of the small number of patients? Have the authors looked into that?
R: We thank the Reviewer for giving us opportunity to clarify this point. In our study, the signatures associated to PTEN, PI3K, and Beta-catenin signaling pathways were not found to be predictive of response, as shown in **Fig 2B**.

9. Similarly, recent papers suggest that cervical tumors harbor HR deficiency <https://www.ncbi.nlm.nih.gov/pmc/articles/PMC8606581/> which usually harbors higher number of T cells. CNV (amplification and deletions together) could indicate genomic instability which are usually linked to higher immunogenicity due to cGAS/STING signaling. Would have been nice to incorporate pSTAT1 and pSTAT3 as correlative markers of type II and type I IFN.

R: We fully agree with the reviewer on the importance on the evaluation of the phosphorylation states of STAT1 and STAT3. STAT1 and STAT3 were not part of the differentially expressed genes of the IFN type I and II pathways (See **Extended Table 2**). However, we do not have remaining tissue for additional IHC assessments.

10. Figure 2: The non-responder patients have signatures of angiogenesis and EMT as well as reduced IFN type I and type II response. These signatures should be properly described with volcano plots of the most differentially expressed genes for responders and non-responders.

R: We thank the reviewer for the suggestion. We have now included an additional volcano plot in **Figure 2D**, highlighting the most differentially expressed genes, within the signaling pathways angiogenesis, EMT (in subjects with progression, PD) and IFN-alpha/gamma (in progression-free subjects, PF). The exhaustive list of the genes is now provided as **Extended Table 2**.

11. Figure 3 and text under “Brisker pre-existing tumor infiltrating immune infiltrate at baseline in responder patients”. Please in the text avoid using terms such as “slight”, “trend”, “difference”, “higher”...replace it by statistically significant or non-statistically significant. Also, and this is valid for all the Figures when the p-value is not statistically significant either add the p-value in numbers or add NS p-value (every column comparison should have a p-value)

R: We thank the Reviewer for this comment. We have revised the text accordingly.

12. Figure 3 mIF images should be of better quality. Epithelial to mesenchymal transition was one of the signatures observed by RNAseq. The authors mentioned that there are differences between stroma and tumor immune infiltration, but I cannot see a specific marker for stroma, was it done?

R: We thank the reviewer for this observation, we will provide a higher resolution version of Figure 3. The distinction between stroma and epithelial compartments was done based on the expression of cytokeratin by mIF. The segmentation between tumor-surrounding stroma (in green) and tumor areas (in red), automatically generated by the software, is shown below. We have now modified the mIF pictures in **Figure 3** to better visualize the stroma-tumor margins.

13. Line 455 Under: “Non-tumoral PD-L1+ cells - comprising immune and stromal cells - significantly accumulated in the stroma of PF patients, along with a significant increase of PD-L1+ tumor cells (expressing cytokeratins, CKs) in the tumor areas of PF patients (Fig. 3D and Extended Data Fig. 2B)

R: We are afraid we do not understand this question.

14. This study's findings will be useful in the design of future clinical trials. For example, this study could help future investigators determine the PD-L1 cut-off as inclusion criteria in future trials. It is thus critical to understand which assay was used to determine PD-L1 staining. Typically, the approved assay for nivolumab clinical trials is antibody 28-8, with a threshold of >1% on TC. Because of the small number of patients and response rates to CRT+N that are very similar to those published in the literature with CRT alone, drawing conclusions will be difficult, but because this is the only information we could eventually have it would be of value if the authors include this in the paper.

R: We thank the reviewer for making this important observation. PD-L1 staining was assessed by immunohistochemistry with the ZR3 clone, that was previously used to evaluate PD-L1 expression in locally advanced cervical cancer following neoadjuvant chemotherapy by Yun Lian *et al*. We could not establish a cut-off for PD-L1 that clearly differentiates PF vs PD patients. A comment has also been included in the Discussion regarding the robustness of the PD-L1 status as biomarker of response to anti-PD(L)1 therapy (**lines 626-628**).

15. There is statistically significant difference in the proportion of CD3 cell in the stroma in responders vs non-responders. Can the authors look at the presence of CD8+PD1+ cells ? These are probably the cells that express GrzB and Ki67+ which can be the effectors (exhausted T cells). It would be important to look at the ratio of CD8+PD1+/CD4+Foxp3+ cells.

R: We thank the reviewer for the suggestion. Due to the limited amount of sample, we could not perform an additional multiplex staining to address this point. However, we did not observe any significant difference when looking at the ratio between n° CD3+GrzB+ / n° CD3+FOXP3+, assessed by mIF, in both tumor and stroma areas, nor the ratio between % CD3+CD8+PD1+ / % CD4+CD25+ as surrogate of Tregs, in baseline tumors assessed by FC.

A) Ratio between CD3+ GzmB+ and CD3+ FOXP3+ cell numbers, assessed by mIF, in the tumor and stroma areas.
B) Ratio between CD8+ PD1+ and Tregs (CD4+ CD25+) in baseline tumors, assessed by FC. Both CD8 and Tregs frequencies were normalized to CD3+ cells.

16. Overall, their findings support the idea that T-cell activation and cytotoxicity can be used to predict anti-PD1 success. Their findings also suggest that preexisting immunogenic and inflamed malignant cells with the ability to generate tumor reactive TILs in situ are required for success. As recent papers show that CD11b+ or CD11c+ cells occupy niches in close proximity to T cells expressing PD1, it is important to understand whether this proximity implies more antigen presentation and co-stimulation capacity from the CD11b+ or CD11c+ cells in cervical tumors, which could also explain more exhausted (PD1+ T cells). Can the authors supply CD80, CD86, and CD28? Due to the lack of CD28 costimulatory cues, it is possible that solitary PD-1+CD8+ TIL are more likely to reach dysfunctional exhausted states in situ. If this is the case, the authors could go back to the DNA sequencing data and look at the signatures that patients with properly costimulated T cells have.

R: We thank the reviewer for this insightful suggestion. To address this question, we performed mIF of CD8, CD28, PD-1, CD11c, CD86 and CK (**Fig.3G,H and Extended Fig.2F**). Due to the limited amount of material and to the number of markers that can be analyzed simultaneously, we could not include CD80 in the panel. As shown in **Fig.3G**, the TME of PF patients is rather enriched in CD11c+ CD86+ antigen-presenting cells (APCs) and CD8+CD28+ (PD-1+ or PD-1-) T cells; however, the quantification of both cell subsets is similar in PF vs PD patients (**Fig.3G,H and Extended Fig.2F**). Interestingly, although not statistically significant, we found that in the tumor-surrounding stroma CD8+ CD28+ PD-1+ T cells were in closer proximity of APCs (CD11c+ CD86+) and of tumor cells in PF vs PD (**Fig.3L**).

17. Recent papers show that the irradiation of the tumor draining lymph nodes abrogates immunological responses. Please see <https://www.nature.com/articles/s41591-022-02084-8>. Supp Fig 7h in this paper: Patient who received 25x1.8 Gy on para-aortal lymph nodes alongside the spine 12 months before the 89ZED88082A PET scan. See also <https://www.nature.com/articles/s41467-022-34676-w> See also <https://pubmed.ncbi.nlm.nih.gov/29898992/>

These papers should be commented as they derive important lessons for the future where we should probably be more economic in the irradiation of lymph node areas where priming takes

place. Particularly knowing that CD8+ T cell activation in cancer comprises an initial activation phase in lymph nodes followed by effector differentiation within the tumor <https://doi.org/10.1016/j.immuni.2022.12.002>

Would the authors propose any solution or hypothesis generating idea in order to tackle this problem?

Could the authors re-analyse the data by separating patients based on para-aortic positive disease or para-aortic irradiation? Did these patients had more immune suppressive signatures at baseline? Do they had more Tregs? More M2 macrophages? How did they respond to treatment? The numbers maybe small, and thus the importance of validating this cohort with other cohorts of patients to derive meaningful conclusions.

R: Thank you for this very insightful comment. We have discussed the detrimental effect of nodal irradiation on the anti-tumor immune response in light of the recent literature. The patients who had para-aortic irradiation didn't have a more immune suppressive signatures at baseline. We observed interesting differences in patients with progressive disease (PD) vs progression-free (PF) ones; indeed, the latter showed a brisker stromal immune infiltrate, higher proximity of tumor-infiltrating CD3⁺ T cells to PD-L1⁺ tumor cells and of Foxp3⁺ T cells to proliferating CD11c⁺ myeloid cells. They also had higher baseline levels of PD-1 on EMRA CD4⁺ T cells and ICOS-L on tumor-associated macrophages (TAMs) vs PD, which instead displayed enhanced PD-L1 expression on TAMs and higher, circulating frequencies of proliferating Tregs at baseline. The PF subjects displayed a more reactive TME, characterized by higher numbers of CD28-expressing T cells at baseline.

Discussion

1. Line 559 "We observed a 25% rate of acute, grade 4 lymphopenia, likely related to a synergistic effect between cisplatin, radiotherapy and nivolumab". I could not find this information in the paper and I found it in the discussion.

Chronic lymphopenia has been described in patients with cervical cancer, correlations analysis using FDG-PET/CT showed that radiation reduces the active bone marrow justifying the chronic myeloid toxicity observed. I would suggest that the authors include their toxicity in the results section and that the lymphopenia is depicted at baseline and at subsequent measurements. In oncology during follow up we perform blood work across several months for these patients, the authors well recognised this in line 567 "These observations should justify repeated laboratory tests". If the authors have the data, it would be important to see baseline and subsequent measurements. For example, it would be important to depict WBC, neutropenia, lymphopenia. This could also enrich Figure 1.

R: We thank the reviewer for the useful suggestion of depicting WBC, neutropenia, and lymphopenia. Baseline and subsequent measurements of WBC, lymphocyte and neutrophil counts are now displayed in **Extended Figure 1B** and this hematological toxicity has also been reported in the result section and in the **Extended Table 1**.

2. The discussion should recognize the short follow-up of the patients which explain why the PFS and LC are so good. Maybe if the data is re-analysed with longer follow-up more non-responding patients will appear.

R: Thank you for raising this important point. We have now mentioned in the discussion that we should be mindful of the short follow-up of the patients in the interpretation of clinical outcome such as PFS (**lines 677-679**).

3. Although the discussion is clinical, the entire paper is translational. The authors should better

discuss their findings and compare them to other papers on gynecological cancers (ovarian, endometrial cancer for instance). Are the TME elements described by the authors universal observations across other tumor types?

R: We thank the reviewer for the insightful suggestion. We have now enriched the discussion with (pre)-clinical studies published in the literature and in reference with the translational data generated in our study.

4. The discussion should recognize as a limitation that biopsies were only obtained at baseline. By analyzing the malignancies before, during and after treatment, investigators could have had a better understanding of the the interaction among the cancer, the treatment and the immune system.

R: Indeed, due to budget limitations, biopsies were only obtained at baseline. In the discussion, we have now mentioned the lack of paired pre- and on-treatment tumour biopsies as a weakness of the study (lines 677-679).

5. Based on the reviewer's new experiments, the investigators could develop proposals for combinations of therapies that will almost certainly be the key to better patient outcomes. Which drugs to use in the future? The problem with immunotherapy today is that its use in patients has outpaced a fundamental understanding of how it works, and we will not reach that understanding unless researchers conduct intensive monitoring of tumors via repeated thoroughly interrogated biopsies.

R: We fully agree with the Reviewer and we believe that future combinatorial modalities testing checkpoint blockade in LACC should enrich for patients harboring tumors with brisk immune infiltrate in proximity of PD-L1⁺ tumor cells, activated, tumor-infiltrating T endowed with costimulatory markers and along with an enrichment in IFN-related pathways.

Reviewer #3 (Remarks to the Author): with expertise in biostatistics, bioinformatics

Comments on the clinical trial study design and statistical analyses.

Rodrigues et al. performed a phase 1 clinical trial of nivolumab plus chemoradiotherapy for locally-advanced cervical cancer. The primary endpoint is the DLT occurrence rate within 11 weeks after the initiation of treatment. The study used a 3+3 design to confirm the dose of nivolumab - 240 mg flat dose q2wk. The dose of nivolumab was initially evaluated in 6 patients and then in additional 9 patients in the expansion cohort. The treatment was considered acceptable if less than 2 patients experiencing DLT in the first 6 DLT evaluable patients.

The 3+3 study design was common in phase 1 studies. The clinical trial data for the primary and secondary endpoints were summarized by descriptive statistics, which are common in reporting phase 1 clinical trial. The DLT rate was 20%, which was considered acceptable.

The patient samples were analyzed by DNA and RNA sequencing. The bioinformatic and statistical analyses are generally appropriate. The major weakness of this report is the small sample size. Among the 16 patients, 12 are responders and 4 are non-responders. The conclusions based on comparison of the 12 responders and 4 non-responders are not convincing due to the small sample size, especially for the non-responders. Some significant findings are found in the responders, but not in the non-responders that might be simply due to the small sample size (e.g., Figure 3 C, D, E, G). In addition, multiple hypotheses were tested for immune markers, but the p-values were not adjusted for multiple comparison. Many marginally significant

comparisons would not be statically significant any more if the raw p-values are adjusted for multiple comparison. The results shown in Figures 3, 4 and 5 should be interpreted carefully.

R: We thank the biostatistician for time evaluation of study and the fully agree with his/her global evaluation. The reviewer raised a valid concern about multiple hypothesis testing without adjusting for multiple comparisons. We consider that our analysis regarding the comparison of immune markers is exploratory in nature and serves as signal for future studies in LACC. The potential for false positive findings is considered in the discussion section by interpreting our findings with caution and nuance and by reporting the limitations of our study. In the discussion, we do recognize that the major weakness of our study is the small sample size. However, we trust that the data can be hypothesis generating and useful for guiding and informing the design of future studies in LACC.

References

1. Cancer Genome Atlas Research Network *et al.* Integrated genomic and molecular characterization of cervical cancer. *Nature* **543**, 378–384 (2015).
2. Scholl, S. *et al.* Clinical and genetic landscape of treatment naive cervical cancer: Alterations in PIK3CA and in epigenetic modulators associated with sub-optimal outcome. *EBioMedicine* **43**, 253–260 (2019).
3. Chakravarthy, A. *et al.* Integrated analysis of cervical squamous cell carcinoma cohorts from three continents reveals conserved subtypes of prognostic significance. *Nat Commun* **13**, 5818 (2022).
4. Chen, M. *et al.* FAT1 inhibits the proliferation and metastasis of cervical cancer cells by binding β -catenin. *Int J Clin Exp Pathol* **12**, 3807-3818 (2019).
5. Kang, S. *et al.* Phosphatidylinositol 3-kinase mutations identified in human cancer are oncogenic. *Proc Natl Acad Sci USA* **102**, 802-807 (2005).
6. Szulzewsky, F. *et al.* YAP1 and its fusion proteins in cancer initiation, progression and therapeutic resistance. *Dev Biol.* **475**, 205-221 (2021).
7. Liang, Y. *et al.* Variation of PD-L1 expression in locally advanced cervical cancer following neoadjuvant chemotherapy. *Diagn Pathol* **15**, 67 (2020).

REVIEWERS' COMMENTS

Reviewer #1 (Remarks to the Author):

Thank you, my concerns have been addressed.

Reviewer #2 (Remarks to the Author):

The authors have replied to all my questions and adapted the manuscript and Figures accordingly